# A way to break bones? The weight of intuitiveness

Delphine Vettese[1,2,3]*, Trajanka Stavrova[1], Antony Borel[1,4], Juan Marín[1,5], Marie-Hélène Moncel[1], Marta Arzarello[2], Camille Daujeard[1]

**1** Histoire Naturelle de l'Homme Préhistorique (HNHP, UMR 7194), Muséum national d'Histoire naturelle (MNHN), Homme et Environnement, Equipe Nomade, CNRS, Institut de Paléontologie Humaine, Sorbonne Universités, Paris, France, **2** Dipartimento degli Studi Umanistici, Sezione di Scienze Preistoriche e Antropologiche, Università degli Studi di Ferrara, Ferrara, Italy, **3** Grupo de I+D+i EVOADAPTA (Evolución Humana y Adaptaciones Económicas y Ecológicas durante la Prehistoria), Dpto. Ciencias Históricas, Universidad de Cantabria, Santander, Spain, **4** Institute of Archaeological Sciences, Eötvös Loránd University, Budapest, Hungary, **5** Institut Català de Paleoecologia Humana i Evolució Social (IPHES), Tarragona, Spain

* delphine.vettese@unife.it

**Data Availability Statement:** Data are available online: Supporting Information 14: Vettese, Delphine; Stavrova, Trajanka; Borel, Antony; Marín, Juan; Moncel, Marie-Hélène; Arzarello, Marta; et al.

## Abstract

During the Paleolithic period, bone marrow extraction was an essential source of fat nutrients for hunter-gatherers especially throughout cold and dry seasons. This is attested by the recurrent findings of percussion marks in osteological material from anthropized archaeological levels. Among them some showed indicators that the marrow extraction process was part of a butchery cultural practice, meaning that the inflicted fracturing gestures and techniques were recurrent, standardized and counter-intuitive. In order to assess the weight of the counter-intuitive factor in the percussion mark pattern distribution, we carried out an experiment that by contrast focuses on the intuitive approach of fracturing bones to extract marrow, involving individual without experience in this activity. We wanted to evaluate the influence of bone morphology and the individuals' behaviour on the distribution of percussion marks. Twelve experimenters broke 120 limb bones, a series of 10 bones per individual. During the experiment, information concerning the fracture of the bones as well as individual behaviour was collected and was subsequently compared to data from the laboratory study of the remains. Then, we applied an innovative GIS (Geographic Information System) method to analyze the distribution of percussion marks to highlight recurrent patterns. Results show that in spite of all the variables there is a high similarity in the distribution of percussion marks which we consider as intuitive patterns. The factor influenced the distribution for the humerus, radius-ulna and tibia series is the bone morphology, while for the femur series individual behaviour seems to have more weight in the distribution. To go further in the subject we need to compare the intuitive models with the distributions of percussion marks registered in fossil assemblages. Thus, it would be possible to propose new hypotheses on butchering practices based on the results presented in this work.

(2020): SI 14: Dataset of observations during the intuitive experiment of long bone breakage. figshare. Dataset. https://doi.org/10.6084/m9. figshare.12709592 Supporting Information 15: Vettese, Delphine; Stavrova, Trajanka (2020): SI 15: Zooarchaeological dataset of observations during the analyses of long bone remains coming from the intuitive experiment of long bone breakage. figshare. Dataset. https://doi.org/10. 6084/m9.figshare.12709709.v1 Supporting Information 17: Vettese, Delphine; Stavrova, Trajanka (2020): SI 17: GIS coordinates of percussion marks for each bone series recorded during zooarchaeological analyses on the remains of long bones after the experiment and cleaning. Skeletal element, individual number, bone number, remain number, percussion marks number, coordinates XY. Percussion mark type are 1: Notch; 2: notch with adhering flake; 3: ovoid pit; 4: triangular pit; 6: crushing mark. figshare. Dataset. https://doi.org/10.6084/m9.figshare.12896852.v1

**Funding:** This projectwas supported by the Fondation Nestlé France (SJ 671–16) (https:// www.nestle.fr/la-fondation-nestle-france); the Centre d'Information des Viandes – Viande, sciences et société (SJ 334–17); and the Muséum national d'Histoire naturelle. The funders had no role in study design, data collection and analysis, decision to publish, or preparation of the manuscript.

**Competing interests:** I have read the journal's policy and the authors of this manuscript have the following competing interests: the authors of this preprint declare that they have no financial conflict of interest with the content of this article. Marta Arzarello and Camille Daujeard are one of the PCI Archaeology recommenders.

## Introduction

Lipids are essential nutrients for the human organism [1–5]. During the Palaeolithic, fat was even more important, due to its role in the gluconeogenesis metabolism, synthesizing glucose from non-carbohydrate precursors [2, 6–8]. At that time, various flora and fauna provided lipid resources but their accessibility was largely influenced by environmental contexts. Lipids were found in oilseeds, such as hazelnuts. However, cold and dry periods were characterized by a low vegetal biomass during which animal resources represented the most important food resources [9–15]. As current population of hunter-gatherer living in cold and dry environment are characterized by hyperproteinic diets [1, 6, 8, 16–19]. The red marrow present in bone epiphyses and yellow marrow from the diaphysis were both important resources of fat from animal carcasses. Marrow is even the most widely available fat resource during the winter [1, 2, 20–24]. Indeed, the recovery of yellow marrow was almost systematic among Palaeolithic hunter-gatherers, and particularly for Neandertals [25–28]. Prehistorians noted widespread evidence of marrow extraction through long bone breakage at Middle and Upper Palaeolithic sites (e.g., [19, 24, 29–41]).

The traces left on bones by marrow recovery are mainly percussion marks. For several decades, percussion marks and long bone breakage methods have been extensively studied (e.g.: [26, 37, 42–45]). Since the beginning of the twentieth century and up until now, archaeological experiments on long bone breakage have been carried out to characterize percussion marks and their location (e.g.: [22, 25, 26, 46–50]). Fracture patterns have also been extensively studied on different states of bone preservation: fresh, dry, weathered, fossilized but also heated, frozen, boiled [51–55]. The authors based their experimental protocols on ethnographic works studying current populations of hunter-gatherers (e.g.: [9, 18, 20, 23, 56–58]). More recently, the use of new methodological approaches (e.g., geometric morphometrics, 3D modelling or GIS analyses of distribution along the diaphysis. . .) presents new challenges for the experimental study of marrow recovery.

Some recent studies have focused on the distribution of percussion marks on long bones to approach the subsistence behaviours of past hominins [19, 59–61]. One of these studies highlighted for the first time a pattern of non-random bone breakage and interpreted it as a product of butchery traditions among Neandertals [59]. Through comparisons with an experiment performed by non-trained individuals, the authors established that this systematic pattern differed from intuitive patterns. In addition, Moclán and colleagues [60] proved that the morphology of the skeletal element from which marrow is extracted could influence the distribution of percussion marks.

We conducted a large-scale experiment to test how non-trained individuals recover marrow. Indeed, in order to identify butchery traditions in Palaeolithic sites, it is necessary to differentiate know-how from intuitiveness, by intuitiveness, in this context, we mean known or perceived by intuition: directly apprehended. The definition of a butchery tradition is a systematic and counterintuitive pattern shared by a same group. The immediate apprehension of the non-trained butcher to break a bone is influenced by numerous variables, including anatomical constraints [59, 60]. Individuals who regularly break long bones acquire an empirical approach and develop specific skills that enhance efficiency [50]. Hence, their skills include habits and preferences gained by experience and/or group traditions. This know-how cannot be assessed without differentiating between physical bone features and socio-cultural practices. In the case of a transmitted systematic practice similar to an intuitive one, it would not be possible to distinguish it from a non-transmitted intuitive practice. So the question of the tradition as a practice in the Palaeolithic can only be evidenced by differentiating it from the intuitive one.

Thus, our aim is to experimentally test whether the inter-individual specificity of long bone morphology may have some influence on the distribution of percussion marks. Based on (GIS) spatial analysis of percussion marks on the bone surface, we also test the existence of a preferential pattern regarding the intuitive breakage of long bones. Moreover, we intend to verify whether non-experienced individuals develop their own method to improve efficiency, such as an auto-learning process. Finally, it should be possible to grasp the influence of each bone structure by comparing the elements between them and by assessing the performance of individuals on the same types of bones. This allows for the comparison of behaviours and the influence of behaviour on the production of bone remains and the marks recorded.

## Methods

### Material

The studied sample includes 120 limb bones (humeri, radio-ulnas, femurs and tibias) from adult cows at least 24 months old. A slaughterhouse supplied this experimental series, with 30 specimens of each element, both left and right Table 1. Professional butchers defleshed the carcasses. During the process, they cut the metapodials with cutting pliers. Thus, these bones are absent from the present experiment. After the reception of the bones, they were stored for less than a week in a fridge at 4˚C. In addition, the elements broken by experimenters 11 and 12 were frozen for 40 days and thawed for three days in the same fridge before the experiment (temperature: 4˚C).

We performed this experiment in a designated area (outside, earthen soil) in the property of the Museum national d'Histoire naturelle (Paris). The experimental series involved 12 individuals without any experience in bone breakage: eight men (mean age = 34 years old; SD = 11.1 years) and four women (mean age = 31 years old; SD = 7.1 years) (Table 2). Five individuals had theoretical knowledge of long bone anatomy and one (individual number 1) was used to flaking lithic tools. Each experimenter broke a series of 10 long bones of the same element. The number of tests is defined by the order in which each bone is broken one after the other in a series of 10. To avoid selection biases, the bones were stacked in a disordered pile when they were presented to the experimenters. Experimenters were isolated from each other so they could not observe how the others broke the bones. No demonstration was performed. Before the experiment, they only received one instruction: break the bone to extract

**Table 1. Data about the bone element broken by each individual (number of ID, element used, side).**

| Individual number | Element | Right | Left |
|---|---|---|---|
| 1 | Humerus | 3 | 7 |
| 2 | Radio-ulna | 6 | 4 |
| 3 | Femur | 6 | 4 |
| 4 | Tibia | 2 | 8 |
| 5 | Femur | 4 | 6 |
| 6 | Tibia | 4 | 6 |
| 7 | Humerus | 5 | 5 |
| 8 | Femur | 6 | 4 |
| 9 | Radio-ulna | 6 | 4 |
| 10 | Tibia | 3 | 7 |
| 11 | Humerus* | 4 | 6 |
| 12 | Radio-ulna* | 3 | 7 |

* These bones were frozen.

**Table 2. Data about the individuals for each series.**

| Individual number | Age | Height (cm) | Weight (kg) | Sex | Used hand | Sport Practice | Bone knowledge | Broken bone element |
|---|---|---|---|---|---|---|---|---|
| 1 | 29 | 167 | 53 | W | Right | Yes | Yes | Humerus |
| 2 | 40 | 174 | 82 | M | Right | No | Yes | Radio-ulna |
| 3 | 25 | 185 | 78 | M | Right | Yes | Yes | Femur |
| 4 | 30 | 186 | 80 | M | Right | Yes | No | Tibia |
| 5 | 51 | 160 | 59 | W | Left | No | Yes | Femur |
| 6 | 22 | 190 | 95 | M | Left | Yes | No | Tibia |
| 7 | 39 | 169 | 63 | M | Right | Yes | No | Humerus |
| 8 | 23 | 182 | 95 | M | Right | No | Yes | Femur |
| 9 | 23 | 167 | 58 | W | Right | Yes | No | Radio-ulna |
| 10 | 46 | 169 | 74 | M | Right | Yes | No | Tibia |
| 11 | 25 | 158 | 73 | W | Right | Yes | No | Humerus |
| 12 | 29 | 178 | 80 | M | Right | Yes | No | Radio-ulna |
| Average | 31.8 | 173.8 | 74.2 | | | | | |
| Stand. dev. | 9.8 | 10.4 | 13.7 | | | | | |
| Maximum | 51 | 190 | 95 | | | | | |
| Minimum | 22 | 158 | 53 | | | | | |

W: Women; M: Men; Stand. Dev.: Standard Deviation; only the individual n°9 have knapping knowledge.

the highest quantity and quality of yellow marrow suitable for consumption. The breakage activity lasted for two to three hours depending on the bone element and the individual. Experimenters had at their disposal a non-retouched quartzite hammerstone weighing about 2 kg, a quadrangular limestone anvil, a plate to deposit the marrow and a wooden stick. The periosteum was not removed before breakage. Experimenters stopped once they had extracted as much bone marrow as they could, and then collected it in the plate and weighed it dx.doi.org/10.17504/protocols.io.be7fjhjn.

Our experiment involved individuals with no empirical knowledge of recovering marrow using long bone breakage. Some were archaeologists or palaeontologists, the rest of the experimenters worked in other fields with no relation to skeletal anatomy or bone tissue properties. The first category of experimenters were familiar with bone morphology and structure as a study object only. Thus, the approach to the long bones during marrow recovery was as intuitive as possible, independently of the experimenters' age or sex. Besides, the experimenters broke a series of ten bones one after another. This process allowed novice experimenters to self-assess and eventually learn from their mistakes.

After breakage of each element, all the fragmented remains were grouped together in a bag with an identification code. The bones were boiled during 2 or 3 hours, to remove the grease and to soften the flesh, and then placed in an oven at 40°C in a solution of water and Papain (papaya enzyme) for 48 hours. Then, they were soaked in a solution of water and sodium perborate for 24 hours, before being air-dried. The material is currently kept in the Institut de Paléontologie Humaine (Paris).

## Data acquisition

During the experiments, an observer recorded the following variables: the series number, element laterality, position of the individual, the way the hammerstone was grasped, the position of the bone, the use of the anvil and the number and location of blows. For these latter, we defined an area as a long bone portion associated with a side (Fig 1). In order to evaluate the

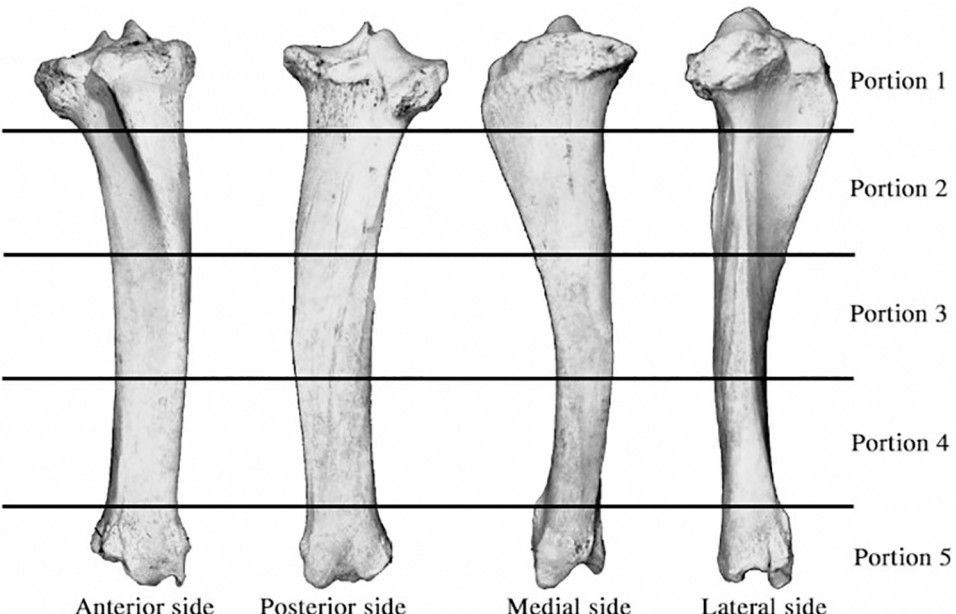

**Fig 1. Bone areas by portions and sides.** Portion 1 (p1): proximal articular end; Portion 2 (p2): proximal diaphysis; Portion 3 (p3): medial diaphysis; Portion 4 (p4): distal diaphysis; Portion 5 (p5): distal articular end and Anterior (a), Posterior (p), Medial (m) and Lateral (l) side (adapted from [62]; https://doi.org/10.1371/journal.pone.0216733.s004).

progression of individuals during the experiment, from the first try to the end of the operation, we recorded task difficulty evaluation and the number of blows. We asked the experimenters to auto-evaluate the difficulty encountered during marrow recovery on a scale from 1 to 5, (1 = very easy; 5 = very hard), considering mainly the opening of the medullary cavity and marrow extraction. This auto-evaluation resulted in an empirical comparison between long bones. The aim of this auto-evaluation was to assess whether individuals progressed and developed an efficient method to extract marrow or if tiredness along with the long period of effort affected their capabilities. For quality control purposes, the experimenter and observer evaluated results on a scale from 1 to 5, 1 referring to contaminated marrow containing abundant splinters and 5 being very clean marrow. The observers are different people with a basic knowledge of bone breakage. It was different from the people who analysed the bones. However, all the breakage sessions were filmed, and the searcher who studied the bone remains watched them carefully to complete when it was necessary the data recorded during the experiments.

The number of identified specimens (NISP), the number of undetermined specimens (NUSP) and the total number of specimens (NSP) were recorded. As the exact location of the percussion marks is necessary, only identified fragments for each bone element were included. Besides, throughout the treatment and the study of each bone, all the remains of one element were kept together, but the NUSP were splinters that we could not refit. Remains that could not be refitted were not considered in the following analysis. The width and length of fragments were measured using a digital calliper. The location of the percussion marks on the bone fragments was analysed according to the GIS protocol developed by Stavrova et al. [62]. The outline of the cortical surface of each identified fragment was digitized using georeferenced photographic images representing the four-sided visualization of each bone as a base.

Bone surface modifications were identified and recorded with the naked eye and 15-20x lens. The centre of each percussion mark was recorded in ArcGIS with a point symbol, based

on the anatomical landmarks of each bone (e.g.,: foramen, crest) [62]. In accordance with the terminology of [63], five types of percussion marks were recorded: adhering flakes, crushing marks, flakes, percussion notches, ovoid or triangular percussion pits and grooves. The number of percussion marks (NPM) and the marks related to percussion (i.e., striations, micro-striations and pseudo-notches) were also noted. These last marks were not taken into account in the total number of percussion marks. In addition, we excluded the flake because if we had taken into account the flake and the notch, which were respectively positive and negative for the same percussion mark, we would have counted the percussion point twice. In this analysis, percussion marks were divided into two groups: 1) crushing marks, notches and adhering flakes (called "CNA") and 2) pits and grooves (called "Pits"). The "CNA" grouped the percussion marks related to cracking and fractures whereas "Pits" were only produced by incipient percussion or the rebound effect.

Distances between percussion marks were calculated in ArcGIS and the "Optimized Hot Spot analysis" tool was used to evaluate and highlight zones with high concentrations of percussion marks. This tool identifies statistically significant spatial clusters of high concentrations (hot spots) and low concentration of percussion marks (cold spots). The number of percussion marks recorded for each series complies with the minimum of 30 features required to conduct the analysis [62].

The data collected during the experiment were compared to the data analysed in the laboratory in order to understand which variables of the experimenters' behaviour could influence fragmentation and the production of percussion marks. Spearman's rho was used to test the correlation between the data collected during the experiment (i.e., number of blows and marrow weight) and the data recorded on the bones after treatment (i.e., number of remains and number of percussion marks). Moreover, in order to evaluate auto-learning and individual improvement, we tested the Spearman correlation of the number of attempts according to the number of blows for each individual. We presented ρ of Spearman. To highlight the evolution of the number of blows between the beginning and end of the experiment for each of the individuals, we compared, on the one hand, the two averages of the first three and the last three trials and, on the other hand, the first five and the last five trials using a non-parametric Wilcoxon test. Indeed, t-test assumptions were not always respected for our data (i.e., the normality and equivalence of the variance of data). We propose an Efficiency Index calculated by dividing the mass of marrow extracted from each bone by the number of blows. This Efficiency Index represents the relationship between a form of expended energy and a form of recovered energy. The higher the Efficiency Index, the more marrow was extracted with the least number of blows. Despite this difference between the elements tested, we observe differences between the bones even in the same individual series: there was a difference in size, in the quantity of spongiosa and in the quantity of marrow. The knowledge of the exact marrow content of each bone was complex to determine before the extraction and varied according the bones. For these reasons, we tested if the fact of intrinsically less accessible marrow having (as for the radius, where the marrow is contained in the proximal diaphysis for example) leads to a multiplication of the number of blows and of remains produced. Our results show that, even if there is on average more marrow contained in the femur, but it is not the bone from which the most marrow has been extracted. While recalling that the experimenters were novices, and some before their first bone, they did not really know where the marrow was and therefore where to place the blows. To test a possible evolution of the Efficiency Index between the three first and the last three tries and also between the five first and the last five attempts, we compared the two averages using the Wilcoxon test, when assumptions for the t-test were not respected. As Cochran's rule was not respected, we used the Fisher exact test to check the independence between the scale of marrow quality of the different elements and for the scale of auto-

assessment. The correlation coefficients and the tests were computed using R software [64]. We use an alpha of 0.05 for all the tests.

# Results

## Data recorded during the experiment

**Bone and individual positions: Different ways to break a bone.** The position of the bone elements and the position of the individuals themselves varied during the breaking process Table 3 (S1 Fig in S1 Appendix). Five different positions chosen by the experimenters were observed. The majority chose the squatting position (45%), followed by the kneeling position (43%). The standing position was rarely selected. Only one individual broke bones in a seated cross-legged position (individual n˚2). A minority of the experimenters adjusted their positions during breakage (five individuals: n˚2, 4, 8, 9 and 11; 42%). Individual n˚7 remained in the squatting position during the entire experiment, but nonetheless tried the standing position once at the eighth try, without repeating it. Individual n˚10 was the only one who did not adopt and maintain a position. By contrast, four experimenters developed some kind of habit after a certain number of attempts. Individual n˚3, after three tries, chose the seated position and individual n˚6 selected a kneeling position after two tries. Individual n˚5 developed a routine from the sixth attempt onwards. For each attempt, she started in a standing position, then kneeled down, and she finished in a squatting position. Individual n˚1 squatted for all the attempts and half the time, she stood to hit the bone on the anvil. Only one individual tested almost all the positions during the experiment (individual n˚12). Nonetheless, after two tries, he switched back to a squatting or kneeling position.

Regarding the choice of hand to grasp the hammerstone, only one individual used the left hand, although two individuals were left-handed and one was ambidextrous. That means that the individual n˚2 used only his right hand only and the n˚5 used simultaneously both hands to hit bones. Therefore, the use of both hands simultaneously did not seem to be dependent on laterality. All the individuals breaking the radio-ulna used both hands at least once to grasp the pebble, which was never the case for the individuals who broke the tibias. The individuals who used both hands were both men and women.

We distinguished five ways to position the long bone in order to open its medullary cavity Table 3. Ten of the individuals used the anvil at least once. Two experimenters (individuals n˚ 8 and n˚11) did not use the anvil at all, but positioned the bones directly on the ground. One individual (individual n˚7) selected the batting technique. He hit the bone directly on the anvil with his hands grasping each epiphysis in order to separate portion 1 from the diaphysis. Then, he laid the largest fragment on the anvil and pursued breakage using the pebble to recover as much marrow as possible. He began using this mixed technique after two tries and repeated it for the next attempts until the end of the series. He judged this mixed technique more efficient than only the hammerstone on anvil technique. One-third (33%) of the individuals always positioned their bones in the same way during the experiment. Therefore, the majority of the experimenters varied bone position during each try. We noticed that individuals experienced some difficulties in stabilizing the bone in order to hit it, especially for the radio-ulnas. Sometimes the bone slipped and the blow was less efficient. Some experimenters used an extra stone, their knees or the anvil to block the bone and keep it from slipping.

**The marrow quantity and quality.** In order to extract marrow after breakage, most of the individuals used the wooden stick we provided, but some preferred selecting their own (S2 Fig in S1 Appendix). Only one individual (n˚7) used the bone splinters created during breakage. Some also used their fingers. The radio-ulna was the long bone that yielded the smallest quantity of marrow; it represents 13% of the total collected marrow (Table 4). The quantity of

**Table 3. Individual and bone position for each bone element.**

| | SQUATTING POSITION | STANDING POSITION | SEATED POSITION | TWO KNEELS POSITION | ONE KNEEL POSITION | RIGHT GRASP HAND | LEFT GRASP HAND | BOTH GRASP HAND | ON THE ANVIL | ONE SIDE ON THE ANVIL | BATTING | ONE SIDE ON ANVIL, OTHER ON THE GROUND | ON THE GROUND |
|---|---|---|---|---|---|---|---|---|---|---|---|---|---|
| **FEMUR** | **7** 23.3% | **4** 13.3% | **10** 33.3% | **13** 43.3% | **2** 6.7% | **20** 66.7% | **0** 0% | **12** 40% | **18** 60% | **0** 0% | **0** 0% | **1** 3.3% | **16** 53.3% |
| 3 | | | 10 | 3 | 2 | 10 | | 2 | 10 | | | | |
| 5 | 6 | 4 | | | | | | | 8 | | | 1 | 6 |
| 8 | 1 | | | 10 | | 10 | | 10 | | | | | 10 |
| **HUMERUS** | **20** 66.7% | **1** 3.3% | **3** 10% | **8** 26.7% | **0** 0% | **30** 100% | **0** 0% | **9** 30% | **20** 66.7% | **3** 10% | **9** 30% | **0** 0% | **10** 33.3% |
| 1 | 9 | 1 | 2 | | | 10 | | | 10 | | | | 10 |
| 7 | 10 | | 1 | | | 10 | | 8 | 10 | 3 | | | |
| 11 | 1 | | | 8 | | 10 | | 1 | | | 9 | | |
| **RADIO-ULNA** | **16** 53.3% | **0** 0% | **11** 36.7% | **5** 16.7% | **3** 10% | **23** 76.7% | **0** 0% | **13** 43.3% | **16** 53.3% | **0** 0% | **0** 0% | **14** 46.7% | **10** 33.3% |
| 2 | | | 10 | | | 10 | | 2 | 10 | | | | 10 |
| 9 | 10 | | 1 | | 3 | 3 | | 1 | 1 | | | 5 | |
| 12 | 6 | | | 5 | | 10 | | 10 | 5 | | | 9 | |
| **TIBIA** | **20** 66.7% | **0** 0% | **0** 0% | **10** 33.3% | **6** 20% | **20** 66.7% | **10** 33.3% | **0** 0% | **26** 86.7% | **2** 6.7% | **0** 0% | **11** 36.7% | **0** 0% |
| 4 | 10 | | | | | 10 | | | 10 | 2 | | | |
| 6 | | | | 10 | | | 10 | | 10 | | | 1 | |
| 10 | 10 | | | | 6 | 10 | | | 6 | | | 10 | |
| **TOTAL** | **63** 52.5% | **5** 4.2% | **24** 20% | **36** 30% | **11** 9.2% | **93** 77.5% | **10** 8.3% | **34** 28.3% | **80** 66.7% | **5** 4.2% | **9** 7.5% | **26** 21.7% | **36** 30.0% |

Percentages were performed by 30 attempts by elements, and by the 120 attempts for the total.

**Table 4. Results of analyses of long bone remains.**

| Element | N°indiv | MW (g) | | NSP | | NISP | | NUSP | | NB | | NPM | | Npit | | NCNA | |
|---|---|---|---|---|---|---|---|---|---|---|---|---|---|---|---|---|---|
| Femur | 3 | 1861 | 10.20% | 259 | 9.50% | 130 | 9.30% | 129 | 9.80% | 641 | 10.30% | 263 | 5.30% | 190 | 4.80% | 73 | 7.40% |
| Femur | 5 | 1030 | 5.60% | 202 | 7.40% | 137 | 9.80% | 65 | 4.90% | 496 | 7.90% | 326 | 6.50% | 282 | 7.10% | 44 | 4.40% |
| Femur | 8 | 1780 | 9.80% | 222 | 8.20% | 113 | 8.10% | 109 | 8.20% | 179 | 2.90% | 221 | 4.40% | 165 | 4.10% | 56 | 5.70% |
| **Femur** | **Total** | **4671** | **25.60%** | **683** | **25.10%** | **380** | **27.10%** | **303** | **22.90%** | **1316** | **21.10%** | **810** | **16.20%** | **637** | **15.90%** | **173** | **17.50%** |
| Humerus | 1 | 1710 | 9.40% | 290 | 10.70% | 109 | 7.80% | 181 | 13.70% | 573 | 9.20% | 194 | 3.90% | 72 | 1.80% | 122 | 12.30% |
| Humerus | 7 | 1737 | 9.50% | 131 | 4.80% | 77 | 5.50% | 54 | 4.10% | 244 | 3.90% | 304 | 6.10% | 217 | 5.40% | 87 | 8.80% |
| Humerus | 11 | 1941 | 10.60% | 195 | 7.20% | 84 | 6.00% | 111 | 8.40% | 377 | 6.00% | 367 | 7.40% | 301 | 7.50% | 66 | 6.70% |
| **Humerus** | **Total** | **5388** | **29.50%** | **616** | **22.60%** | **270** | **19.30%** | **346** | **26.20%** | **1194** | **19.10%** | **865** | **17.30%** | **590** | **14.80%** | **275** | **27.80%** |
| Radio-ulna | 2 | 799 | 4.40% | 263 | 9.70% | 119 | 8.50% | 144 | 10.90% | 808 | 12.90% | 180 | 3.60% | 49 | 1.20% | 131 | 13.20% |
| Radio-ulna | 9 | 775 | 4.20% | 113 | 4.20% | 80 | 5.70% | 33 | 2.50% | 873 | 14.00% | 503 | 10.10% | 428 | 10.70% | 75 | 7.60% |
| Radio-ulna | 12 | 811 | 4.40% | 189 | 6.90% | 73 | 5.20% | 116 | 8.80% | 822 | 13.20% | 942 | 18.90% | 894 | 22.40% | 48 | 4.80% |
| **Radio-ulna** | **Total** | **2385** | **13.10%** | **565** | **20.80%** | **272** | **19.40%** | **293** | **22.20%** | **2503** | **40.10%** | **1625** | **32.60%** | **1371** | **34.30%** | **254** | **25.70%** |
| Tibia | 4 | 1755 | 9.60% | 439 | 16.10% | 192 | 13.70% | 247 | 18.70% | 285 | 4.60% | 280 | 5.60% | 138 | 3.50% | 144 | 14.50% |
| Tibia | 6 | 1759 | 9.60% | 175 | 6.40% | 120 | 8.60% | 55 | 4.20% | 753 | 12.10% | 738 | 14.80% | 673 | 16.80% | 65 | 6.60% |
| Tibia | 10 | 2290 | 12.50% | 244 | 9.00% | 166 | 11.90% | 78 | 5.90% | 189 | 3.00% | 669 | 13.40% | 590 | 14.80% | 79 | 8.00% |
| **Tibia** | **Total** | **5804** | **31.80%** | **858** | **31.50%** | **478** | **34.10%** | **380** | **28.70%** | **1227** | **19.70%** | **1687** | **33.80%** | **1401** | **35.00%** | **288** | **29.10%** |
| | *Total* | *18248* | | *2722* | | *1400* | | *1322* | | *6240* | | *4987* | | *3999* | | *990* | |

Individual number (n°indiv), Marrow Weight (MW), Number of Specimen (NSP), Number of identified specimens (NISP), Number of undetermined specimens (NUSP), Number of blows (NB), Number of percussion marks (NPM), Number of pit (Npit) and Number of crushing marks, of adhering flakes and of notches (NCNA).

marrow extracted from the femur is twice as high (26%). The individuals who broke the humeri and the tibias extracted a slightly higher quantity of marrow (around 30%). The total quantity of yellow marrow recovered represented 18 kg. We did not notice high variation among individuals, except for individual n°5 who recovered a significantly smaller quantity of marrow from the femurs than the other individuals did for the femur series (S3-S6 Figs in S1 Appendix).

The evaluation of the quality of collected marrow shows a low proportion of bad or very bad assessments (10%) (Table 5). In addition, most of those who made this judgment broke radio-ulnas. Most (2/3) of the recovered marrow was evaluated as good or very good to

**Table 5. Scales of difficulty felt during the experiment and of the marrow quality auto-evaluated by the experimenters themselves for all the elements and for each one.**

| Marrow quality scale | All | | Femur | | Humerus | | Radio-ulna | | Tibia | |
|---|---|---|---|---|---|---|---|---|---|---|
| Very bad | 3 | 2.50% | 0 | 0% | 3 | 10% | 0 | 0% | 0 | 0.00% |
| Bad | 9 | 7.50% | 1 | 3.30% | 0 | 0% | 8 | 26.70% | 0 | 0.00% |
| Neutral | 29 | 24.20% | 7 | 23% | 15 | 50% | 6 | 20% | 1 | 3.30% |
| Good | 55 | 45.80% | 17 | 56.70% | 12 | 40% | 14 | 46.70% | 12 | 40% |
| Very good | 24 | 20% | 5 | 16.70% | 0 | 0% | 2 | 6.70% | 17 | 57% |
| **Felt scale** | **All** | | **Femur** | | **Humerus** | | **Radio-ulna** | | **Tibia** | |
| Very easy | 14 | 11.70% | 4 | 13.30% | 2 | 6.70% | 1 | 3.30% | 7 | 23.30% |
| Easy | 53 | 44.20% | 11 | 36.70% | 21 | 70% | 5 | 16.70% | 16 | 53.30% |
| Moderate | 28 | 23.30% | 9 | 30% | 4 | 13.30% | 11 | 36.70% | 4 | 13.30% |
| Hard | 16 | 13.30% | 4 | 13.30% | 1 | 3.30% | 8 | 26.70% | 3 | 10% |
| Very hard | 9 | 7.50% | 2 | 6.70% | 2 | 6.70% | 5 | 16.70% | 0 | 0% |

consume. None of the individuals who broke the humerus series evaluated the marrow as very consumable. However, more than a half of the individuals estimated the tibia marrow to be very good. The Fisher exact tests show that the scales of the tested elements were independent (df = 12, p<0.05).

We noted a significant negative correlation between the quantity of marrow and the number of blows. Indeed, the radio-ulna was the element that received the most blows during the experiment and where a smaller quantity of marrow was recovered. Concerning the radio-ulna, we observed a significant negative correlation between the NISP and the quantity of marrow collected. Besides, we noticed a significant positive correlation between the quantity of recovered marrow and both the NUSP and the number of CNA.

**The level of difficulty experienced during the experiment and the number of blows.** In conclusion, the experimenters found the activity very easy 14 times (12%), easy 53 times (44%), moderate 28 times (23%), difficult 16 times (13%) and very difficult only 9 times (13%). individuals who broke the humeri did not encounter any difficulty in most cases (N = 92; 77% easy or very easy). The radio-ulna seems to be more difficult to break (N = 52; 43% hard or very hard and N = 44; 37% moderate). None of the individuals who broke a tibia estimated the task to be very difficult Table 5). They did not find the task any easier or harder as they progressed. In other words, they did not find the task harder at the beginning because they did not know how to go about it and then easier because they found an efficient technique. Likewise, exhaustion from the activity did not influence whether individuals experienced more difficulty at the end of the breakage series (S3-S5 Figs in S1 Appendix). The difficulty encountered according to the bone element varied significantly (Fisher exact test: df = 12, p<0.01).

The results of Spearman's correlation analysis between the number of attempts according to the number of blows for each individual completed these observations. For five individuals (individuals n°1 and n°7 (humerus), individual n°3 (femur) and (individuals n°6 and n°9 (tibia)), the correlation analysis shows a significant negative relationship (Spearman's rank correlation coefficient results respectively ρ = -0.81; -0.77; -0.73; -0.93; -0.69). Regarding the other individual series, the results were not significant (Spearman's rank correlation coefficient results respectively ρ: n°2 = -0.29; n°4 = 0.54; n°5 = -0.2; n°8 = 0.51; n°10 = 0.41; n°11 = -0.44; n°12 = 0.32).

The results of Spearman's correlation analysis between the number of attempts according to the number of blows for each individual completed these observations. For five individuals (individuals n°1 and n°7 (humerus), individual n°3 (femur) and individuals n°6 and n°9 (tibia)), the correlation analysis shows a significant negative relationship (Spearman's rank correlation coefficient results respectively ρ = -0.81; -0.77; -0.73; -0.93; -0.69). Regarding the other individual series, the results were not significant (Spearman's rank correlation coefficient results respectively ρ: n°2 = -0.29; n°4 = 0.54; n°5 = -0.2; n°8 = 0.51; n°10 = 0.41; n°11 = -0.44; n°12 = 0.32).

The proportion of the number of blows varied over the last attempts depending on individuals. Experimenters n° 1, 3, 7, 9 reduced their number of hits by half, on average, n°6 by six between the three first attempts and the last three (S3 Fig in S1 Appendix). Individual n°12 increased the average number of blows after the three first tries. However, if we consider the same individual breaking the same element during the whole experiment, we did not observe a significant difference between the five first and the last five attempts. We found similar results between the three first and the last three, except for the humerus (p-value = 0.0254) (S4 Fig in S1 Appendix).

The bi-plots showing the number of blows and marrow weight did not display any linearity in the data regarding each element (S5 Fig in S1 Appendix). When we examine bone element

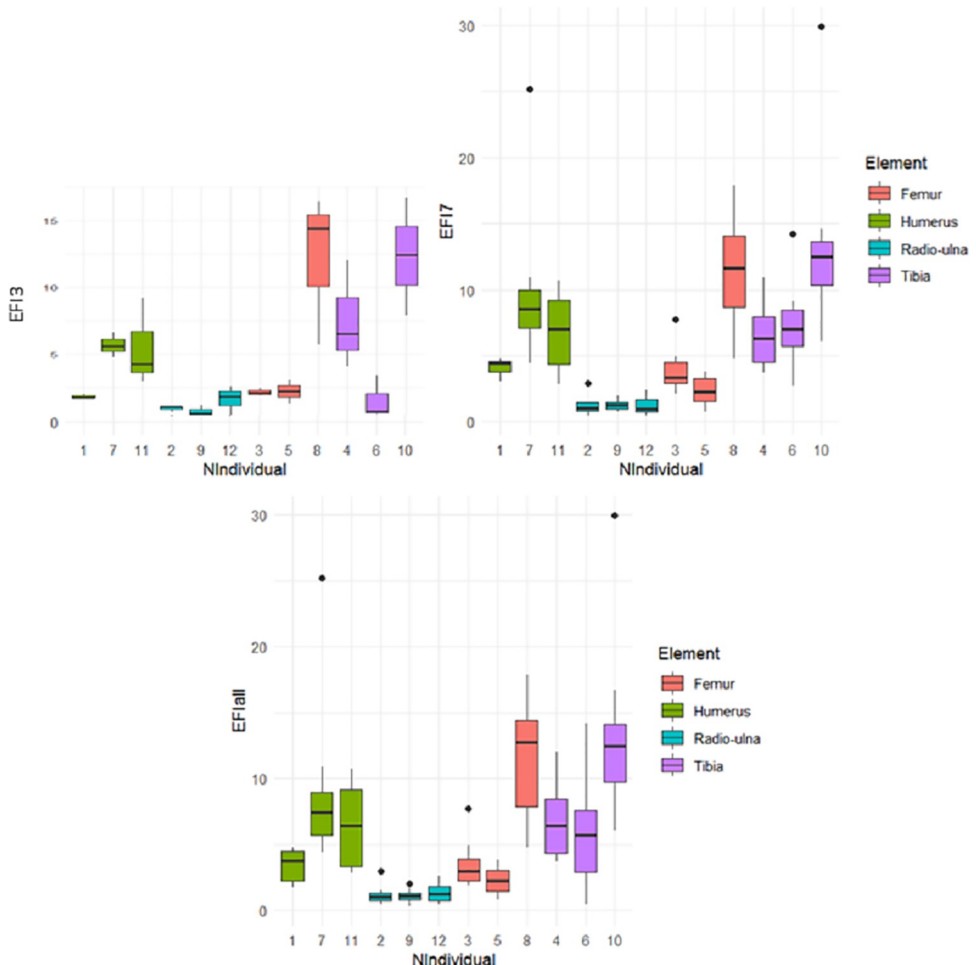

**Fig 2. Box plots of the Efficiency Index.** Efficiency Index by Individual during the experiment regarding the third first tries, the last seventh tries and all tries.

type, the individuals who broke the radio-ulnas reveal reduced Efficiency Index compared to the other elements (Fig 2; EFI all). On the other hand, the experimenters who broke the tibias had a high average Efficient Index (>5). There was marked variation between the individuals who broke the humeri and the femurs, especially for the first three attempts (Fig 2; EFI 3). Between the first three and the last seven tries, the EFI of the individuals who broke the tibias varied, like individual n˚8. At the individual level, two-thirds of the individuals increased their EFI between the first five and last five tries (n˚3, 5, 1, 7, 11, 9, 6, 10). Similarly, more than half the individuals augmented their EFI between the first and last three (n˚3, 1, 7, 11, 2, 9, 6) (S6 Fig and S1 Table in S1 Appendix). However, regarding all the experimenters, the Wilcoxon signed rank test did not show a significant difference between the first five and the last five tries or between the first three and the last three tries (p-value > 0.05) (S2 Table in S1 Appendix). We observed similar results regarding each element (p-value >0.05).

## Data recorded on the faunal assemblage obtained

**Bone fragmentation.** The experiments yielded 2,722 bone remains, among which 1,400 were refitted Table 4. The NSP of the tibias (32%) is higher than for the other elements, which

are almost identical (humerus: 25%, radio-ulnas: 23% and femurs: 21%). Likewise, the tibia NISP is the highest (34%) and the NISP of the humerus and radio-ulna are the lowest (19%). Nonetheless, we noticed that the average NUSP is around 25% for all the elements. We did not observe marked variability between individuals regarding the NISP, NUSP and the NSP SI 6). However, individual n˚4 produced more numerous NUSP, which also affected the NSP. According to our application of the Spearman's rank correlation test, the NSP was positively correlated with both the NUSP and the NISP for all the elements (all elements: $\rho = 0.44$ and $\rho = 0.76$; humerus: $\rho = 0.57$ and $\rho = 0.79$; radio-ulna: $\rho = 0.39$ and $\rho = 0.68$; femur: $\rho = 0.51$ and $\rho = 0.8$; tibia: $\rho = 0.56$ and $\rho = 0.84$). The NUSP and the NISP were also positively correlated between them (all elements: $\rho = 0.91$; humerus: $\rho = 0.93$; radio-ulna: $\rho = 0.91$; femur: $\rho = 0.9$; tibia: $\rho = 0.91$) (S7 Fig in S1 Appendix).

**Number of percussion marks.**   For one-third of the series, we recorded more percussion marks than the number of blows given by the experimenters Table 4. This anomaly was observed for one individual for each bone element. Three of the individuals who produced a high number of percussion marks used an anvil, but this was not the case for individual n˚8. The tibia and radio-ulna series presented numerous percussion marks (around 33%), corresponding to almost twice as many percussion marks as the femurs and humeri. We recorded the most abundant pits and grooves on the tibia and the radio-ulna series. The other percussion marks represented around 27% for all elements, with slightly less on the femurs (18%). Furthermore, these marks display a lower standard deviation (SD = 33) than the pits (SD = 263) or all the percussion marks combined (SD = 245). We observed an inter-individual difference, in particular for the first series of the humeri, radio-ulnas and tibias. A different observer recorded these series. However, both researchers applied the same protocol and each checked their records.

According to our application of the Spearman's rank correlation test, we noted a significant positive correlation between percussion marks with the number of pits and grooves on one hand, and the number of crushing marks, adhering flakes and notches ("CNA"), on the other, for all the long bones ($\rho_{NPM/Npit}$: humerus = 0.95; radio-ulna = 0.96; femur = 0.95; tibia = 0.87; all elements = 0.93; $\rho_{NPM/NCAN}$: humerus = 0.35; radio-ulna = 0.7; femur = 0.35; tibia = 0.43; all elements = 0.48). The number of pits and grooves in the tibia series depended on the number of percussion marks, whereas the other marks were independent. The number of percussion marks causing a fracture ("CNA") was not significantly correlated with the other percussion marks (pits) ($\rho_{NCAN/Npit}$: humerus = 0.16; radio-ulna = 0.59; femur = 0.29; tibia = 0.47; all elements = 0.24). However, we observed a positive correlation for the radio-ulna series. The number of percussion marks producing fractures ("CNA") was positively correlated with the NSP for the humerus series ($\rho = 0.34$). We observed a significant positive correlation between the "CNA" percussion marks and the weight of marrow and the NUSP for femurs (respectively, femurs: $\rho = 0.05$). In addition, the number of pits and grooves on the tibias was correlated positively with the NISP and the NSP (respectively: $\rho = 0.5$; $\rho = 0.38$). Finally, the radio-ulnas showed a positive correlation between percussion marks causing breakage ("CNA") and the number of blows ($\rho = 0.39$) (SI 9). We noted also a significant negative correlation between the number of percussion marks and the weight of marrow for the tibia series ($\rho = -0.59$).

The graph for the PCA analyses showed more differences between the tibia series on one hand and the radio-ulna and humerus series, on the other, in particular on the second dimension (Fig 3). These differences are mainly due to the NISP and the Efficiency Index. On the first dimension, radio-ulnas and humeri are opposed by the number of percussion marks and by the number of tries.

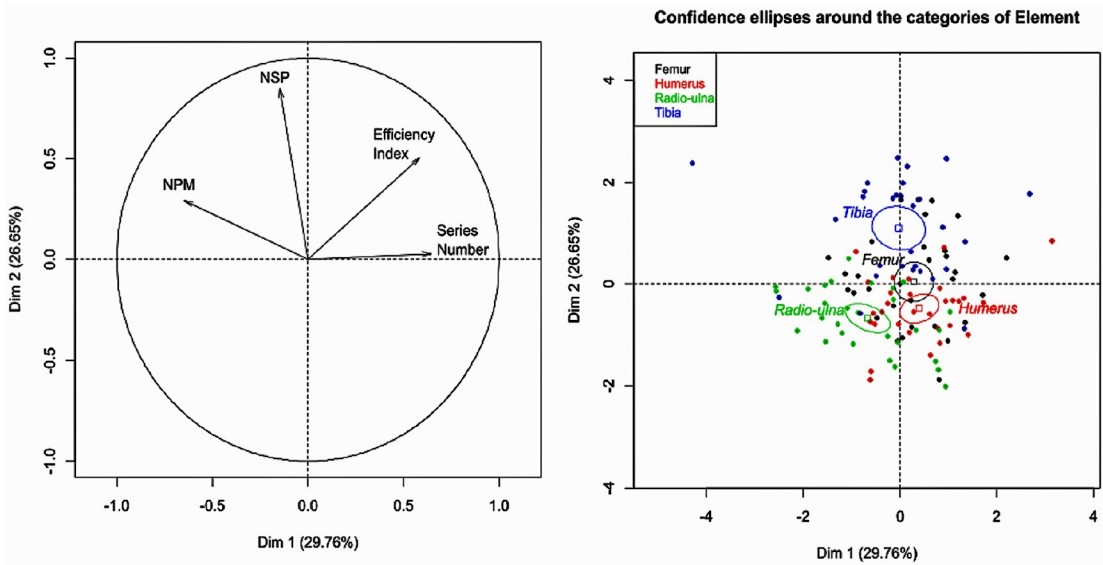

**Fig 3. PCA analyses of variable tested.** PCA regarding the number of identified remains (NISP), the number of percussion marks (NPM), the Efficiency Index and the number of attempts (Series Number) with ellipses at 0.95.

**Percussion mark types and spatial distribution.** We observed around 5,000 percussion marks on the identified fragments Table 6. Pits were the most numerous (78%) and grooves were the least frequent marks (2.6%). Both triangular and ovoid pits were present, but the former were less numerous. We recorded more pits on the tibias and radio-ulnas than on the other bone types. Adhering flakes and notches represented 15% of all the percussion marks.

**Table 6. Type of percussion marks by individual and by element on NISP.**

| Element/N °individual | Percussion notch | | Adhering flake | | Triangular pit | | Ovoid pit | | Grooves | | Crushing marks | | Total PM |
|---|---|---|---|---|---|---|---|---|---|---|---|---|---|
| **Femur** | **85** | **10.5%** | **26** | **3.2%** | **41** | **5.1%** | **574** | **70.9%** | **22** | **2.7%** | **62** | **7.7%** | **810** |
| 3 | 45 | 17.1% | 10 | 3.8% | 9 | 3.4% | 173 | 65.8% | 8 | 3.0% | 18 | 6.8% | 263 |
| 5 | 15 | 4.6% | 5 | 1.5% | 14 | 4.3% | 257 | 78.8% | 11 | 3.4% | 24 | 7.4% | 326 |
| 8 | 25 | 11.3% | 11 | 5.0% | 18 | 8.1% | 144 | 65.2% | 3 | 1.4% | 20 | 9.0% | 221 |
| **Humerus** | **80** | **9.2%** | **93** | **10.8%** | **95** | **11.0%** | **462** | **53.4%** | **33** | **3.8%** | **102** | **11.8%** | **865** |
| 1 | 39 | 20.1% | 66 | 34.0% | 8 | 4.1% | 47 | 24.2% | 17 | 8.8% | 17 | 8.8% | 194 |
| 7 | 21 | 6.9% | 18 | 5.9% | 44 | 14.5% | 166 | 54.6% | 7 | 2.3% | 48 | 15.8% | 304 |
| 11 | 20 | 5.4% | 9 | 2.5% | 43 | 11.7% | 249 | 67.8% | 9 | 2.5% | 37 | 10.1% | 367 |
| **Radio-ulna** | **120** | **7.4%** | **77** | **4.7%** | **625** | **38.5%** | **717** | **44.1%** | **29** | **1.8%** | **57** | **3.5%** | **1625** |
| 2 | 66 | 36.7% | 39 | 21.7% | 9 | 5.0% | 33 | 18.3% | 7 | 3.9% | 26 | 14.4% | 180 |
| 9 | 39 | 7.8% | 17 | 3.4% | 141 | 28.0% | 275 | 54.7% | 12 | 2.4% | 19 | 3.8% | 503 |
| 12 | 15 | 1.6% | 21 | 2.2% | 475 | 50.4% | 409 | 43.4% | 10 | 1.1% | 12 | 1.3% | 942 |
| **Tibia** | **174** | **10.3%** | **86** | **5.1%** | **411** | **24.4%** | **943** | **55.9%** | **47** | **2.8%** | **24** | **1.4%** | **1687** |
| 4 | 93 | 33.2% | 42 | 15.0% | 26 | 9.3% | 93 | 33.2% | 19 | 6.8% | 5 | 1.8% | 280 |
| 6 | 23 | 3.1% | 31 | 4.2% | 224 | 30.4% | 435 | 58.9% | 14 | 1.9% | 11 | 1.5% | 738 |
| 10 | 58 | 8.7% | 13 | 1.9% | 161 | 24.1% | 415 | 62.0% | 14 | 2.1% | 8 | 1.2% | 669 |
| **Total** | **459** | **9.2%** | **282** | **5.7%** | **1172** | **23.5%** | **2696** | **54.1%** | **131** | **2.6%** | **245** | **4.9%** | **4987** |

Total PM: Percussion marks total.

Crushing marks were one of the least numerous traces (4.9%), and were often located close to the articular portions. We recorded fewer crushing marks and more grooves on tibias compared to the other long bones. Conversely, we documented the most numerous crushing marks and adhering flakes on the humeri. We noted high variation in the number of pits and adhering flakes depending on the observers who documented the percussion marks. This difference in recording did not influence the number of other marks to the same degree. We also observed inter-individual variability, in particular for the radio-ulna series. Besides, the bone frozen presented mixed fracture planes [52]. However, there does not seem to have any role on the distribution of impact marks and the percussion marks typology. The 20 elements (10 humerus and 10 radio-ulnas) that have been frozen do not show a divergent distribution pattern compared to those that have not been frozen.

The crushing marks were often situated close to the articular portions (Fig 4, S8-S11 Figs in S1 Appendix). We documented some crushing marks on the medial diaphysis of the humeri and the radio-ulnas and the tibias of individual n˚4. Furthermore, we recorded most of the percussion marks on the diaphysis; only the pits and grooves were located on the articular portions. It was difficult to observe a concentrated area of percussion marks on the template on which we drew the marks. We merged all the percussion marks derived from the 10 bone elements into a single series and performed an optimized hot spot analysis to evaluate high concentration zones of percussion marks (Figs 5, 7, 9, 11). For each long bone series, we tested all the percussion marks together; apart from percussion marks causing a fracture ("CNA") and the others (pits and grooves) (Figs 6, 8, 10, 12).

The analyses taking into consideration all the percussion marks generally highlighted cold spots on the articular portions, for all the studied long bones. Conversely, hot spots were generally highlighted on shaft portions. In particular, most of the time the ulna was a cold spot for all the series of radio-ulnas. Likewise, for almost all the series, the tibia crest was a cold spot. The majority of the cold spots were observed on proximal and distal articular portions with the exception of the proximal portion of the radius in the "CNA" analyses and on the tibia crest and ulnas (Fig 4, S8-S11 Figs in S1 Appendix).

The spatial analyses show that the majority of the high confidence hot spot zones for the humerus series were on the medial and lateral sides (Fig 5A, 5B and 5D). Individual n˚11 was an exception with hot spot areas on portion 4 of the posterior and anterior sides. The merged humerus series showed four high confidence hot spot zones on the medial and lateral sides. The hot spot zone on the medial side of portion 2 is the largest, and the one on the lateral side of portion 2 is the smallest. The last ones are located on portion 4 (Fig 5A–5D). When we consider the two types of percussion marks, we observe some differences in the location of the high confidence hot spot zone. The "CNA" were on the lateral and medial sides of portion 2, the pits and grooves were also on the medial side of portion 2, but also on all sides of portion 4 (Fig 6).

For the radio-ulna series, we observed hot spot areas on the anterior and posterior sides for all the individuals (Fig 7). Individual n˚9 was the only one who frequently hit portion 3–4 on the medial side. The analysis of the combined radius-ulna series highlighted a high confidence hot spot zone on the posterior side of diaphysis portions 2 and 3. This result reflected the general tendencies observed for all individuals. The analysis defined a reduced area, portion 2 on the anterior side, as a hot spot zone, which was influenced by individual n˚9. The results of the analysis showed that the high confidence hot spot zone for the merged radio-ulna was on portion 3 of the posterior side, and on the anterior side portion 3–4 for "CNA" marks only (Fig 8D).

The spatial analyses of the femur series show diversified distribution of the hot spot zones along the diaphysis, depending on the experimenter (Fig 9). For the three series, all sides were

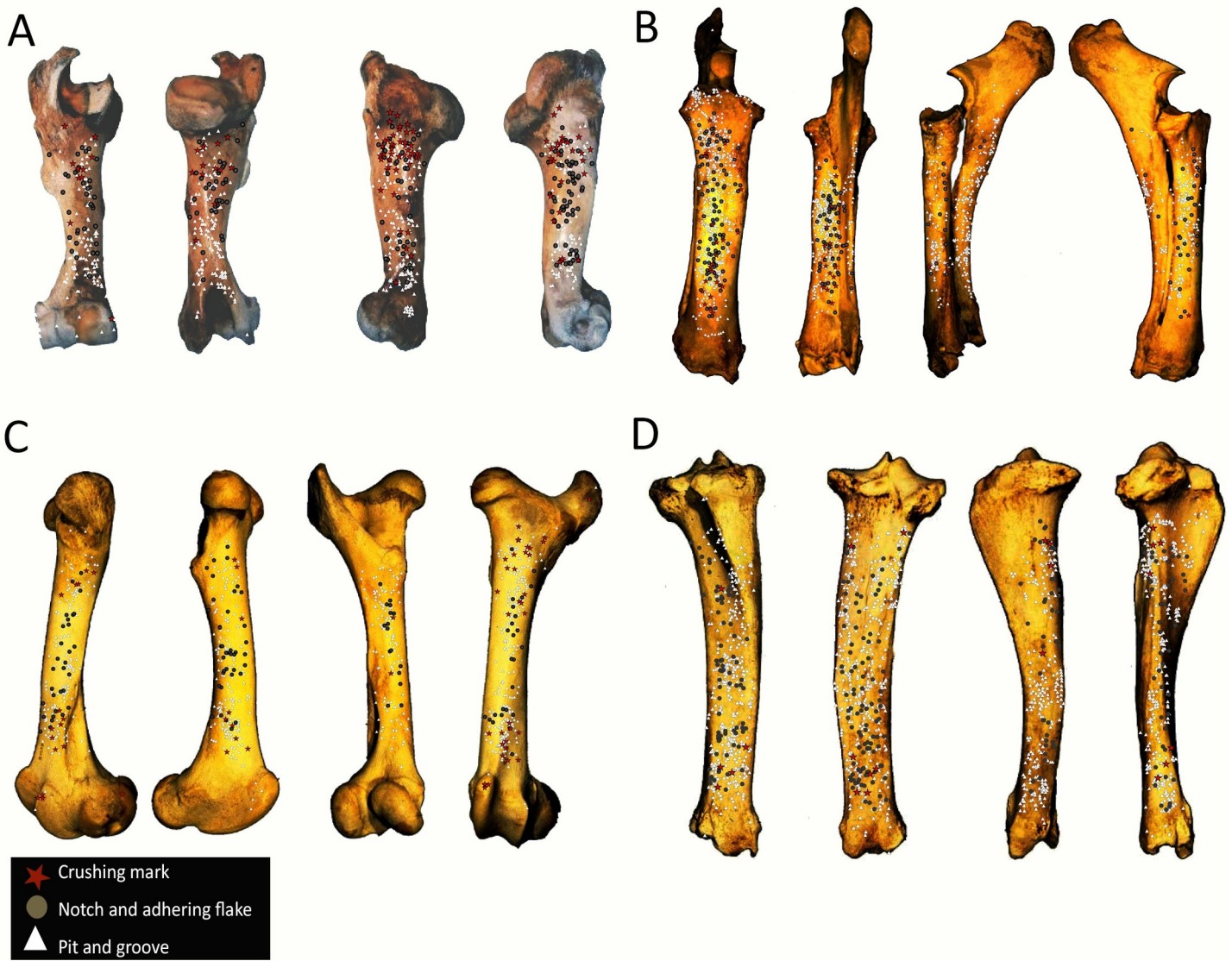

**Fig 4. Distribution of percussion marks of individual series merged along the humerus, the radio-ulnas, femurs and tibias.** The type of percussion mark are red star: crushing marks, white triangle: pits and grooves and grey circle: notches and adhering flakes.

a hot spot area at least once, reflecting high inter-individual variability. However, the spatial analysis of the merged femur series shows three confident hot spot zones, quite similar to individual n˚8 (Fig 9C), on portions 2 and 4 on the anterior side, and on portions 3 and 4 on the medial side. High confidence hot spot zones for both percussion mark types are more numerous and dispersed on the merged femur than on the other bones (Fig 10D and 10H).

For the tibia series, we observed high confidence hot spot zones on anterior and posterior sides for each individual, in particular on portions 3 and 4 (Fig 11). The lateral side highlighted hot spot areas for individuals n˚6 and n˚10, especially on portion 2 (Fig 11B and 11C). The merged tibia series reflects the dispersion of the high confidence hot spot zones observed for each series. The spatial analyses showed six hot spot zones, two of which very limited in size. However, the "CNA" area of high confidence hot spot was only on the posterior side of portions 3 and 4 (Fig 12A–12D). Zones with pits and grooves were numerous, distributed on the diaphysis of the lateral, anterior and posterior sides (Fig 12E–12H).

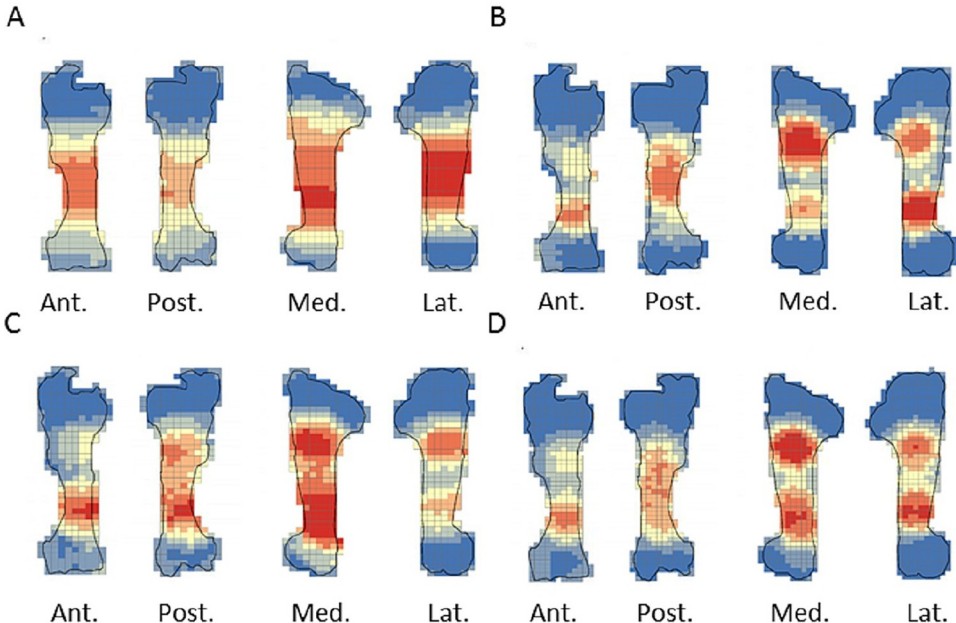

**Fig 5. Optimized hot spot analysis of combined percussion marks from the humerus in each series.** A- Individual n˚1; B- Individual n˚7; C- Individual n˚11; D- Merged humerus series (Ind.: Individual).

## Discussion

The most significant information highlighted by our experiments is related to the variability observed during the breakage process and that observed among the bone elements produced during experimentation. Indeed, some individual behavioural tendencies emerge during the breakage process, as well as patterns in the production and distribution of percussion marks. Our results also brought to light the influence of long bone morphology in an intuitiveness context.

### How do individual gesture alternatives and bone elements influence breakage methods?

Some individuals could apply reduced force when they hit the bones, especially during the first attempts. Initially, experimenters tested the resistance of the bones and some were afraid of hurting themselves. In these cases, they hit the bones with less force. Likewise, at the end of the experiment, tiredness could result in a reduction of the force applied. After the activity, most of the individuals had a shaking hand, and all the individuals had aches and pains the following days. This shows that marrow extraction requires a good physical condition and/or practice and/or good technical skill. During the activity, some individuals progressed or developed habits or a routine. However, they did not seem to become more efficient at optimally breaking bone to extract the maximum quantity of marrow. In addition, during the experiment, the individuals mentioned inter-element differences, whereby some bones were easier to break than others, irrespective of the experience acquired or the degree of exhaustion. Moreover, it is important to note that sometimes the hammerstone slipped on the cortical surface or slipped out of the experimenter's hand. This happened especially when the hammerstone and hands were covered in grease after several tries. When an individual broke several bones, he/she had to clean the tools or his/her hands to pursue marrow extraction.

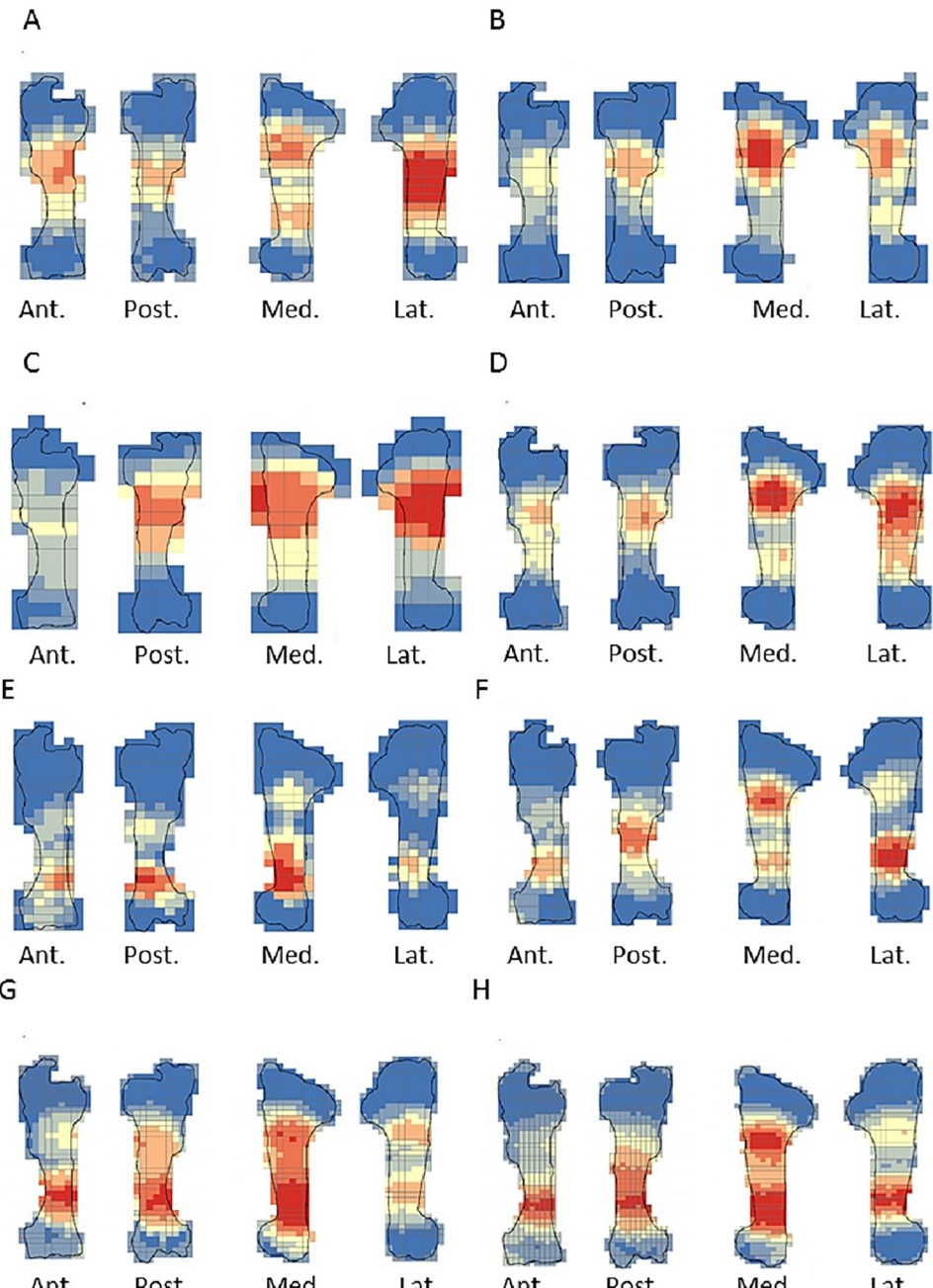

**Fig 6. Optimized hot spot analysis of combined percussion marks from the humerus in each series.** A-E-Individual n˚1; B-F- Individual n˚7; C-G- Individual n˚11; D-H- Merged humerus series; A, B, C and D: "CNA" Percussion marks and E, F, G and H: pits and grooves (Ind.: Individual).

One of the most significant results of our experiment was the variability of individual behaviour during marrow recovery. One individual adopted an unexpected posture: a cross-legged seated position. However, the observed variability only comprised five positions. In addition to these postural variations, we documented some non-linear behaviour during the experiment. Our experiment showed that non-trained individuals could change their position at different moments during both the marrow extraction from one bone and throughout the

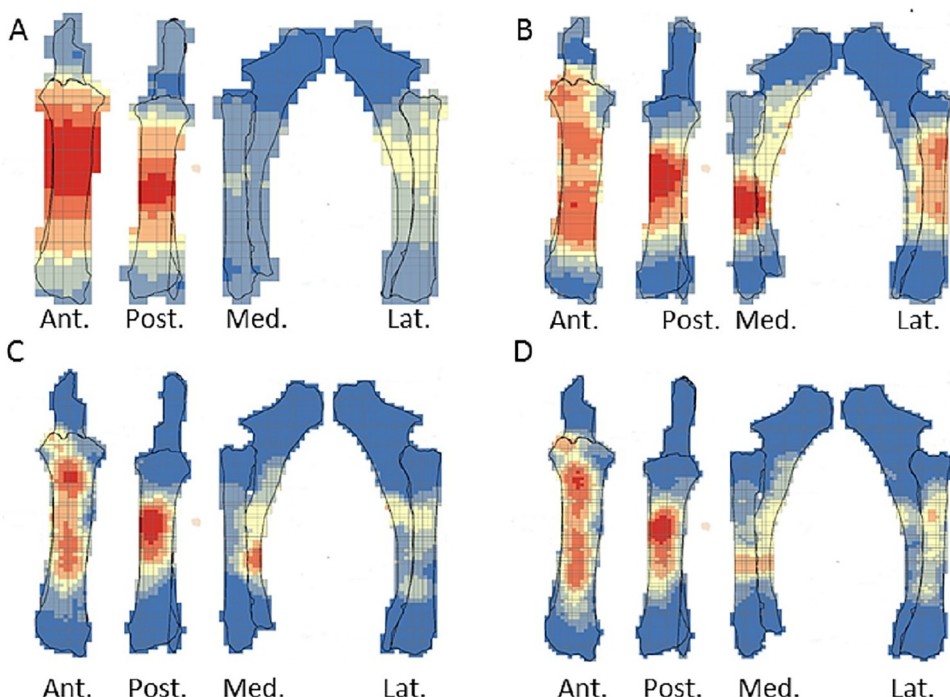

**Fig 7. Optimized hot spot analysis of combined percussion marks from the radio-ulnas in each series.** A-
Individual n˚2; B- Individual n˚9; C- Individual n˚12; D- Merged radio-ulnas series (Ind.: Individual).

ten bone series. However, the majority of the individuals kept the same position or developed some kind of habit or routine in their practice. This routine concerned the individual and bone position and the way of grasping the bone and percussor. The auto-learning developed by experimenters to be as efficient as possible involved their position and their way of grasping the bone and percussor. The position influences the amplitude of the gestures, the force involved in breakage and the location of the blows. The standing position was only selected by women, and not by all of them. Sex did not influence the choice of posture or position of the bone. No link was found between the gender of the individual and how consistent they were in their posture choice or in positioning the bone throughout the experiment. Likewise, sex did not seem to influence the way the percussor was grasped or the use of an anvil. These results could provide some evidence regarding the organization of butchery task distribution during the Palaeolithic. We noted that the posture adopted by individuals did not vary in relation to the skeletal element. However, the type of long bones seemed to affect practices and gestures, in particular for tibias and radio-ulnas. This observation could be related to the high number of blows applied to these two bone types by novice experimenters.

Although individuals proceeded differently depending on the elements, we noticed some divergences in our results, in particular between radio-ulnas and tibias. The experimenters who broke the radio-ulna series had a very low yield in relation to the number of blows. On the whole, the opposite is true for the tibia series, for which individuals had a high yield. Thus, individuals who broke radio-ulnas found the activity quite difficult and considered the marrow to be of relatively poor quality. On the other hand, the individual who broke tibias found the activity quite easy and the extracted marrow was considered to be of relatively good quality. Furthermore, experimenters used two hands at least once for breaking radio-ulnas, but never for tibias. The use of both hands to grasp the pebble could be an expression of the difficulty felt

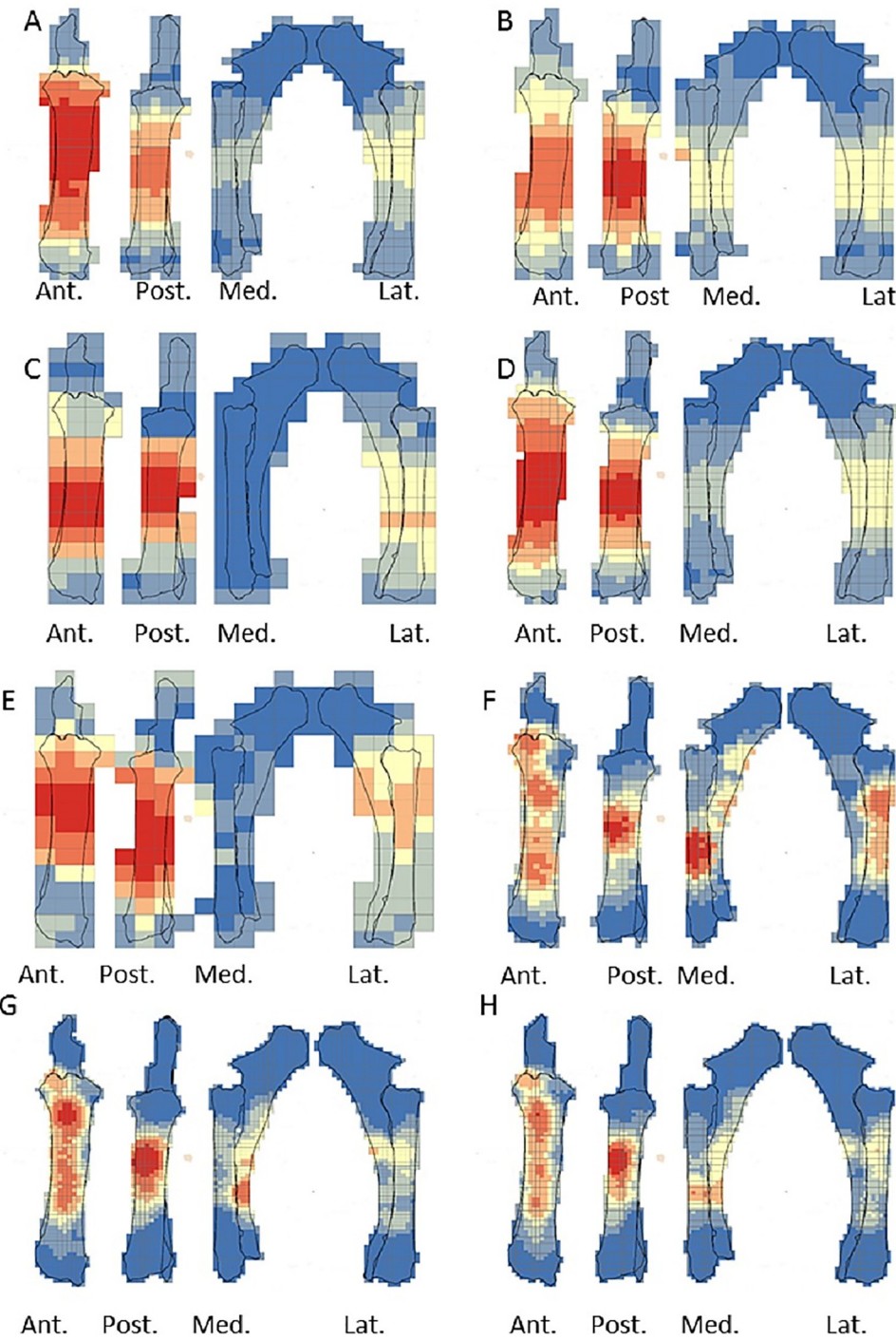

**Fig 8. Optimized hot spot analysis of combined percussion marks from the radio-ulnas in each series.** A-E-Individual n˚2; B-F- Individual n˚9; C-G- Individual n˚12; D-H- Merged radio-ulnas series; A, B, C and D: "CNA" Percussion marks and E, F, G and H: pits and grooves (Ind.: Individual).

by the individuals breaking radio-ulnas. Regarding the NSP produced for both these elements, we recorded the highest number of blows and the lowest NSP for the radio-ulnas and the exact opposite for the tibias of our sample. This was why we observed a high number of pits and the

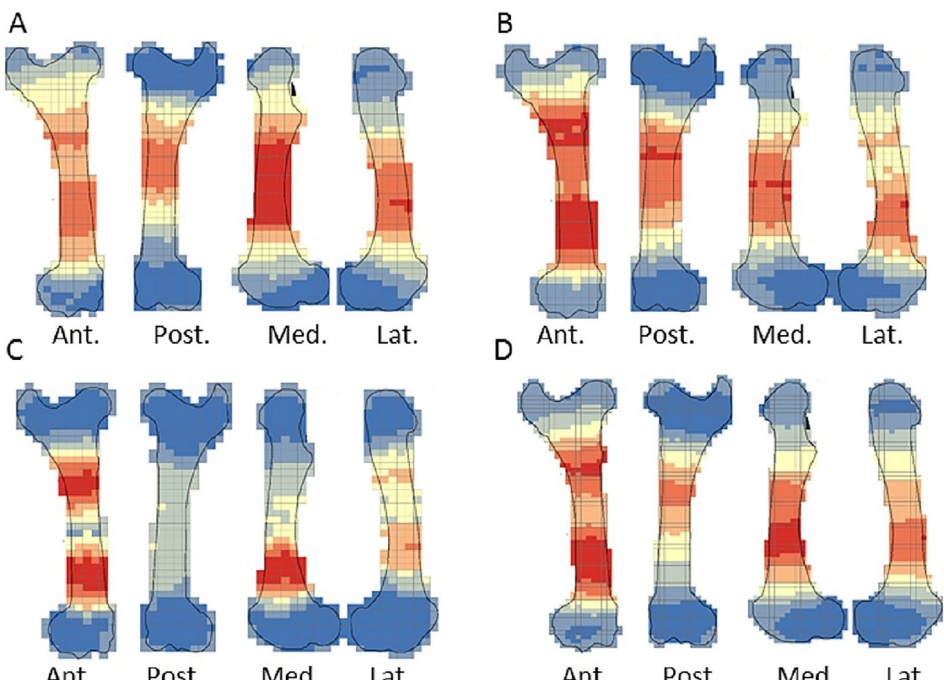

**Fig 9. Optimized hot spot analysis of combined percussion marks from the femurs in each series.** A- Individual n˚ 3; B- Individual n˚5; C- Individual n˚8; D- Merged femurs series (Ind.: Individual).

lowest number of "CNA" percussion marks on radio-ulnas. Contrary to the tibia, the breakage of a radio-ulna of an animal such as a cow requires some mastery. Numerous studies have highlighted differences between the volumes of the medullar cavity of these two long bones. Tibias and femurs comprise large medullary cavities. On the other hand, medullar cavities are smaller for radio-ulnas and humeri and thus contain a lesser quantity of yellow marrow [9, 21, 65]. Therefore, the EFI (Efficiency Index) highlighted differences between the energy spent by counting the blows and the quantity of marrow extracted, in part because of the difference in volume between the radio-ulna and tibia series. However, medullary cavity volume alone cannot account for all the differences observed between bones. In terms of density, tibias and radiuses show the highest diaphysis density of all the long bones used in our experiment [66, 67]. However, they seem to react differently to blows. This difference was highlighted by the PCA, showing an opposition between the radio-ulna and the tibia series regarding the NSP, the Efficiency Index and the number of percussion marks.

## What correlation can be made between blows and percussion marks?

Various researchers have stressed that the aim of mastering percussion breakage was to reduce the number of blows required to remove all the marrow from the medullary cavity [43, 48, 60]. For this reason, it was expected that novice experimenters would average more blows than experts. Indeed, we noted a high number of blows, ranging from 6–279, with an average of 52, for the entire bone experiment. If we compare our experiment with previous works, we observe a huge difference between the average number and range of blows Table 7. The average number of blows by individuals in our experiment was higher than the maximum in other studies. In particular, for their first tries, two individuals hit the bones more than 200 times. Besides, none of the experimenters stopped hitting the bones after the first fracture. The presence of the periosteum on the bones was not sufficient to explain the very high number of

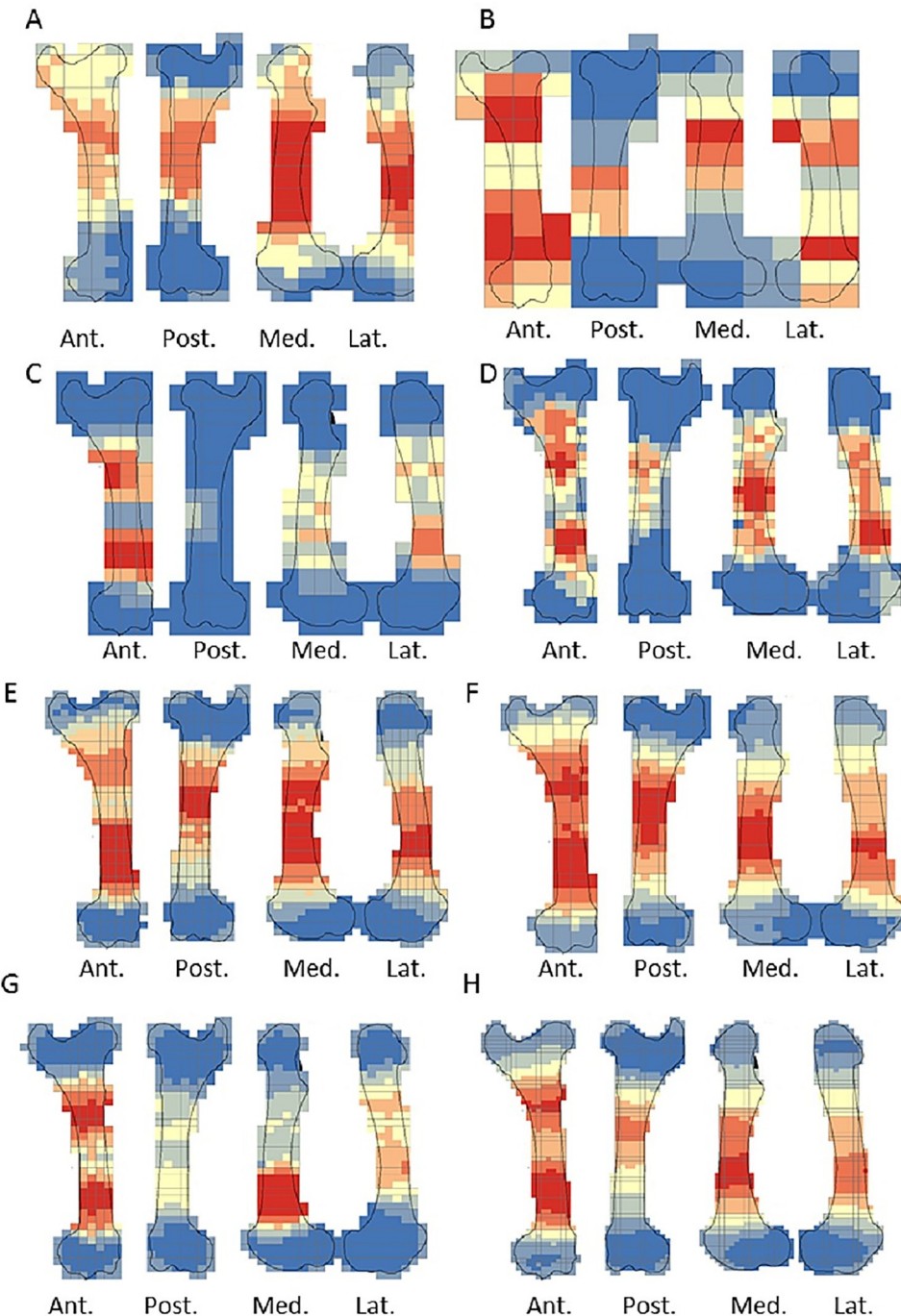

**Fig 10. Optimized hot spot analysis of combined percussion marks from the femurs in each series.** A-E- Individual n˚3; B-F- Individual n˚5; C-G- Individual n˚8; D-H- Merged femurs series; A, B, C and D: "CNA" Percussion marks and E, F, G and H: pits and grooves (Ind.: Individual).

recorded blows. Indeed, during some experiments quoted in Table 7, the periosteum was not removed [49, 50, 68]. Novice experimenters tried to break the element further in order to recover the maximum amount of marrow. The weighing of the marrow after each attempt motivated some of them. At the individual scale, we observed some increase or decrease in the

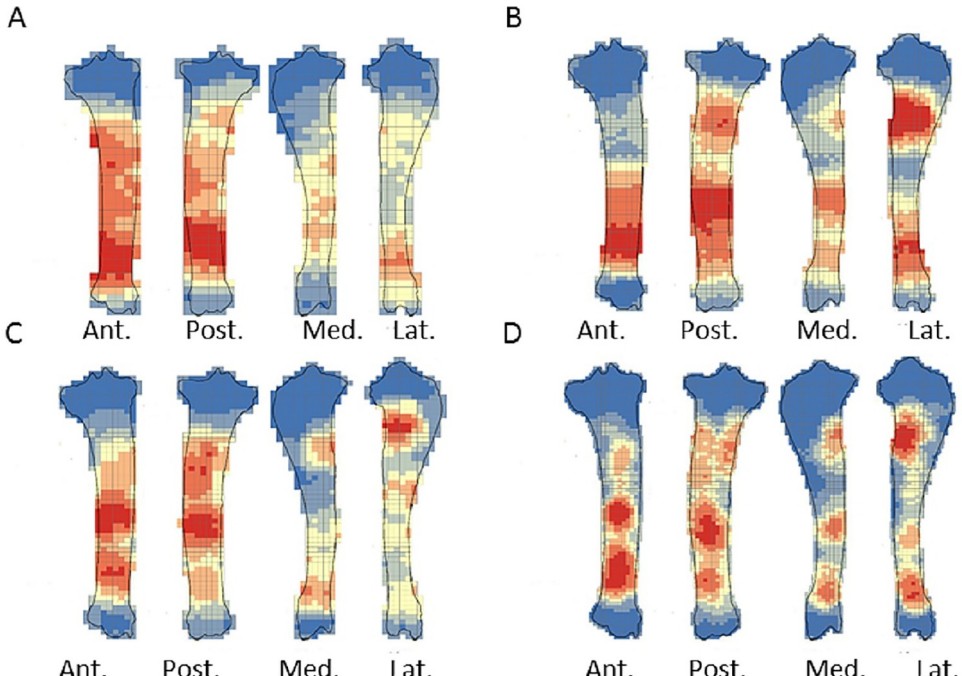

**Fig 11. Optimized hot spot analysis of combined percussion marks from the tibias in each series.** A- Individual n˚ 4; B- individual n˚6; C- Individual n˚10; D- Merged tibias series(Ind.: Individual).

number of blows at the beginning and the end of the experiments, but overall, we did not highlight significant progress for all the individuals or a particular element. The only exception was the humerus, for which the number of blows decreased for all the individuals. The number of blows and the individuals' decision to pursue breakage after the first fracture can increase the NSP. Intensive breakage can reflect a high number of blows in order to access the yellow marrow. In addition, we observed the conservation of almost all the epiphyses in their entirety. Indeed, it is not necessary to break the articular portions for the extraction of yellow marrow only.

Interestingly, the number of blows is not correlated with the number of percussion marks. Comparing the two values, we noted that for two-thirds of the individual bone elements, the number of blows is higher than the number of percussion marks and more surprisingly, for one-third of the bone elements, the number of percussion marks is higher than the number of blows. The surprise does not come from the count, but from the fact the blow do not every time create a percussion mark, so the multiplicity of the percussion marks are not directly linked to the impact point. The counter blow due to the anvil could create percussion marks. Besides, one blow could create more than one percussion mark due to the irregularity of the hammerstone, even if it is not modified. But this need to be further explored. For one individual per bone element (humerus: n˚7, radio-ulna: n˚12, femur: n˚8 and tibia: n˚10), the number of blows was inferior to the number of percussion marks.

In most cases (more than 95%), the number of pits and grooves or "CNA" marks was different to the number of hits. In an archaeological context, it can thus be impossible to infer the number of blows from the number of percussion marks. The use of an anvil or the hammerstone could explain the differences recorded between the number of marks and blows. In their experiment, [50] also observed a majority of pits among the recorded percussion marks. They concluded that most of these pits were produced by the counterblow of the anvil. In our

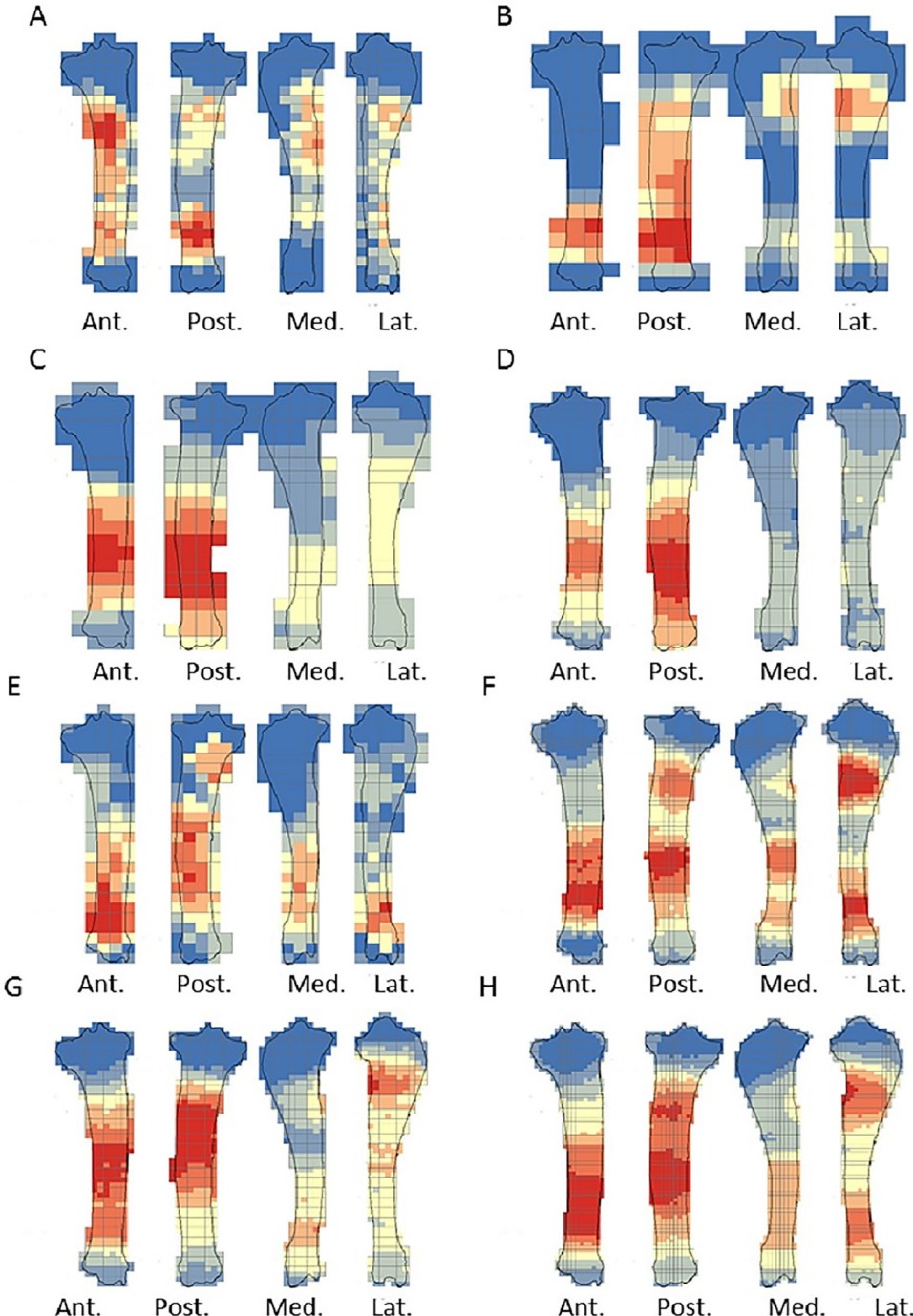

**Fig 12. Optimized hot spot analysis of combined percussion marks from the tibias in each series.** A-E- Individual n˚4; B-F- Individual n˚6; C-G- Individual n˚10; D-H- Merged tibias series; A, B, C and D: "CNA" Percussion marks and E, F, G and H: pits and grooves (Ind.: Individual).

experiment, pits were generally more numerous than the other marks (67%). Thus, pits and groves result from the rebound effect caused by the hammerstone or by the anvil due to the counterblow. In this last case, the counterblow is not a mark; but rather the result of the rebound effect of the bone on the anvil. The marks due to counterblows include pits,

**Table 7. Listing of different experiment of bone breakage for marrow extraction and comparing data.**

| References | Species | Element | Range of number of blows | Mean of number of blows | NSP/ NME = AverageNSP |
|---|---|---|---|---|---|
| [68] | Horse | Femurs | 1–27 | 8.8 | - |
| [48] | Cow | Humerus | - | - | 145/22 = 6.5 |
| | | Femurs | - | - | |
| | | Radio-ulna | - | - | |
| [50] | White deer | Radius | 2–15 | 6 | 811/38 = 21.3 |
| | | Humerus | 1–4 | 2 | 472/36 = 13.1 |
| [69] | Cow | Femurs | 1–8 | 3.5 | 189/13 = 14.53 |
| [70] | Cow | Limb bones | - | 20 | - |
| [49] | Horse | Femurs | 1–3 | 2 | - |
| | | Tibia | 3 | 3 | |

NME: Number of Minimum Element.

microstriations, outer conchoidal scars or notches (e.g.,: [42, 43, 45, 55, 66, 71]), located on the opposite side to the blow. Due to the multiplicity of traces and the overlapping of the traces produced by the hammerstone or when the long bone is batted against a hard surface, it is difficult to discriminate blow marks from counterblow marks. In addition, we recorded some triangular pits and grooves even though experimenters only used non-retouched pebbles. These marks, characterized by straight and sharp edges, could be the product of the rebound effect from the anvil. Therefore, the use of an anvil could create supplementary marks during the blow that is the counterblow. The shape of the hammerstone, which is round but could have irregularities, could also produce many pits. Two individuals (n°8 and 11) did not use the anvil at all, but simply laid the bones on the ground. Nonetheless, some triangular pits and grooves were identified (around 10%). However, in this case too we sometimes recorded more marks than blows. Thus, the counterblow is not the sole explanation for the difference between the number of blows and the number of percussion marks. It is possible that the shape of the hammerstone, the impact with the ground or the collision of bones between them during the activity created several pits. More than one-quarter of the recorded pits were very small and superficial, with a diameter of 10 mm or less (28%). In an archaeological context, the different taphonomical processes, such as trampling, for example [72], can erase these pits or produce others. For the remaining two-thirds of the broken elements, we recorded a lower number of percussion marks than blows. In our protocol, the periosteum was never removed. The presence of the periosteum, which absorbed some of the force of the blows, could partly explain this difference.

## Percussion marks distribution

The areas where the least percussion marks were observed, apart from the articular portions, were the anterior crest of the tibia and the diaphysis of the ulna. The distribution of percussion marks showed that novice experimenters avoided shape constraints, such as bone protuberances. They seemed to prefer relatively flat areas. Our results show that, there is no impact of freeze on the percussion marks distribution. Moreover, during the experiment all the bones were completely defrosted. Thus, the spatial analyses for tibias and radio-ulnas highlighted significant hot spot zones for the merged series, where bone shape was the flattest. Furthermore, for the "CNA" marks, a single hot spot area was most impacted on the tibias. This area was

located on the posterior side of the distal diaphysis. It was one of the flattest and smoothest zones of the tibias with no muscular lines. Moreover, the distal diaphysis is the least dense portion of the tibia (index density = 44) [66]. Regarding the "CNA" marks, two specific high confidence hot spot zones were identified on the radio-ulnas. They were located on relatively flat and easily accessible areas to blows. The first one is situated on the anterior side of the medial shaft, which is relatively flat considering the globally very arched shape of the radio-ulna. The radio-ulna section in this area is flattened compared with portions 3 and 4, which present a semi-lunar crescent shape. When the bone was placed on a small anvil, one of the best ways to stabilize the radio-ulna was to place the medial shaft on the posterior side. Thus, the anterior side was exposed and very accessible. The other hot spot area was on the exact opposite side because, on the ground, it was easier to place the bone on the anterior side to avoid interference from the ulna. In addition, when the radio-ulna was positioned on the anterior side, this area was located at the inflexion point of the bone curve and was the most accessible to hit. Furthermore, these areas were located on portions 2 and 3, which are the densest part of the radius (marrow index = 0.56 & 0.62).

Unlike zeugopods, the diaphyseal section of humeri and femurs is rather cylindrical. Their diaphyses are relatively straight in comparison to the radio-ulna. The articular portion shows important differences in terms of stability. The humerus is more stable on its lateral and medial sides whereas the femur is relatively stable on all sides. Indeed, the very rounded shape of the humerus head causes instability when placed on its posterior side. This is why the distribution of percussion marks, highlighted by the hot spot analyses, is concentrated on the lateral and medial sides. Pit and groove marks are generally distributed on the distal part of the bone. This part of the humerus diaphysis is denser than the others (index density = 0.48). Conversely, the hot spot area of "CNA" marks is only located on the proximal shaft portion, in the least dense part of the bone (index density = 0.25). These "CNA" marks represent one of the most the numerous hot spot zones. In addition, these hot spot areas were similar for the three merged spatial analyses of the femurs (all percussion marks, "CNA" marks and pits and groove marks). All the diaphysis portions were hot spot zones at least once, independently of density variability.

Thus, percussion mark location seems to be highly influenced by the morphological constraints of bones, in particular for the radio-ulna, the tibia and the humerus, in an intuitive context. Individuals were faced with bone marrow extraction for the first time and seem to be highly influenced by shape constraints. These constraints were so strong that individuals continued hitting these zones after many attempts, as bone stability and the accessibility of these areas facilitated breakage and therefore marrow recovery. Nonetheless, the absence of morphological constraints seems to highlight novice choices. Therefore, based on the results of our analyses, in an intuitive context, femurs seem to be the bones for which individual variability was highest.

## Archaeological, experimental and ethnographical data comparisons: Some convergences

Binford [9] described the practices and gestures of the Nunamiut in order to infer past hunter-gatherer behaviour from archaeological assemblages. His observations describe systematic breakage with the recording of the location of the percussion marks after hitting always the same bone portions. Moreover, in the same way as experiments, ethnographical studies highlight wide-ranging variables, which are hardly accessible in archaeological contexts.

We highlighted the existence of a preferential pattern for the intuitive breakage of long bones, using the spatial analysis of the location of percussion marks on radio-ulnas, tibias and

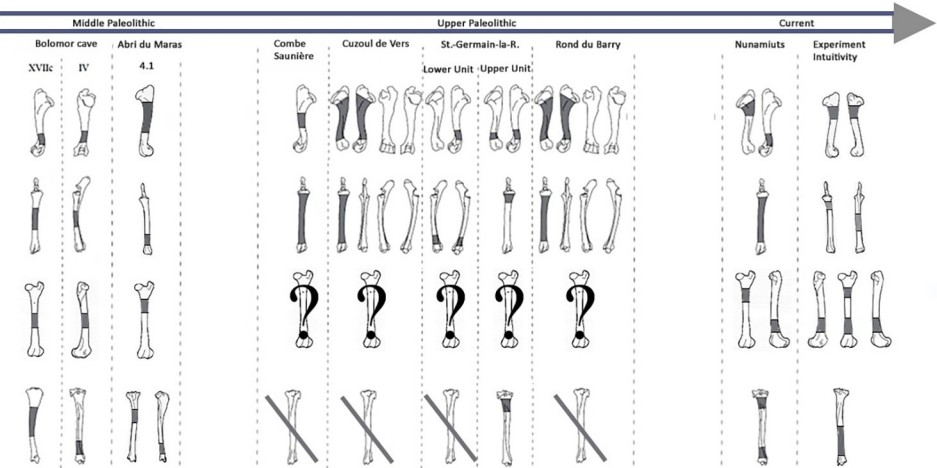

**Fig 13. Location of percussion marks (grey areas) for the humerus, radio-ulna, femur, and tibia for sites from the Middle Paleolithic.** Sites: Bolomor cave (ungulates: Blasco et al., 2013.) and Abri du Maras (middle sized-ungulates: [61]); the Upper Paleolithic: Combe Saunière (reindeer, Solutrean: Castel, 1999), Cuzoul de Vers (reindeer, Badegoulian: [73]), Saint-Germain-la-Rivière, Lower Unit (saiga antelope, Lower Magdalenian: [74]) and Upper Unit (saiga antelope: [19]), Rond du Barry (ibex, Upper Magdalenian: [74]), for these series the data regarding the femurs were missing and most of the tibias did not present higher tendencies; Current: Nunamiut people of Alaska (reindeer: [9] and the results of our experiment regarding the CNA). Adapted to [19].

humeri. Archaeological and ethnographical studies focusing on the distribution of percussion marks along long bone diaphyses have revealed the existence of preferentially impacted areas [9, 19, 59–61]. The comparison between these results and ours underlines the influence of bone shape on some systematic practices of extant and past hunter-gatherers (Fig 13). For radio-ulnas, archaeologists record percussion marks on the anterior side, in particular in the faunal assemblages of level XVIIc of Bolomor Cave [59] and at the Abri du Maras layer 4.1 [61], where some tendencies were observed (Middle Palaeolithic), Combe Saunière and Cuzoul de Vers [73] and the "upper unit" of Saint-Germain-la-Rivière [19, 74] and the Rond du Barry (Upper Palaeolithic), and also among the extant Nunamiut [9]. The areas highlighted by hot spot analyses were similar to the areas of percussion mark concentrations in these comparative examples. These faunal assemblages represent practically all the Palaeolithic sites for which percussion mark distribution has been studied. The faunal assemblages are dominated by different species: reindeer, saiga antelope, ibex, red deer or larger-sized ungulates, like horse or bison, and in our experiment, cow. This comparison suggests that the high shape constraints of the radio-ulnas strongly influenced the distribution of percussion marks on whole long bone remains in an archaeological level independently of ungulate size. Furthermore, for ungulates, the fact that the radius is merged with the ulna constrains the morphological shape of the bones. However, level IV of Bolomor Cave and the "lower unit" of Saint-Germain-la-Rivière showed different patterns on the lateral side for the former site and both lateral and medial sides for the latter (Fig 13).

For the humerus, the distribution of percussion marks was quite similar to our observations. Almost all the sites presented in the work of Masset et al. [19] reveal a pattern on the medial and/or lateral sides, with the sole exception of Bolomor Cave level IV. Compared to the other sites, we noted variability in the exact area impacted by percussion marks, but we observed a preference for portion 4 in the majority of the assemblages. A minority of the studied tibia series showed a systematic pattern or strong tendencies. The posterior sides displayed the most areas of percussion mark concentration. Finally, most of the femur series from the

archaeological sites studied do not seem to show systematic distribution, with the exception of level VI of Bolomor, which indicates two preferentially impacted zones, and the Abri du Maras, where one area was more impacted than the other. We observe similarities with the Nunamiut for the anterior side.

## Conclusion

For the purposes of testing intuitiveness in bone breakage, we experimentally tested the influence of individual behaviour and long bone morphology on percussion mark location. We observed a high variability of gestures during the breakage process. Most of the individuals developed their own routine. This routine included several positions and different techniques, involving variability in the applied force and different bone element responses. Nevertheless, our results seem highlighted similarities in percussion mark distribution despite the specificity of each experimenter, for most of the long bones studied here. Indeed, one of the most important results is the identification of specific hot spot areas, regardless of the variability of experimenters' behaviour during the breakage process. These similarities could be considered as intuitive patterns.

The location of percussion marks is influenced by numerous variables depending on: the experimenters' behaviour and skills, the techniques employed, the location of the blows, the morphology of the skeletal element and bone position during breakage. Nevertheless, our results seem to highlight the predominance of two main factors in an intuitive context: long bone morphology for the humerus, radio-ulna and tibia and experimenters' behaviour for the femur.

From a different perspective, we have shown that bone response could be very different in an intuitive context, notably for the radio-ulna and the tibia. These differences could be observed at almost all the different levels of the analysed data: experimenter behaviour or perception, marrow quality and quantity, number of blows and remains produced and type of percussion marks recorded. These results highlight the diametrically opposed differences between tibias and radio-ulnas in an intuitive context. Besides, to go further and specify our results, more experiment will be required with various cooking methods, like roasting or boiling bones. Furthermore, the effect of prior periosteum removal will be tested regarding the percussion marks formation or their number, and also, the breakage efficiency.

Our results shed new light on the importance of gestures in the breakage process. It could prove interesting to model these results and evaluate their amplitude and the force involved. Numerous studies focus on the force required to break long bones but generally concentrate on pressure methods. Consequently, data pertaining to the percussion process are lacking. In addition to measuring the force applied by the blow, it is essential to evaluate the force of the counter-blow caused by different materials, such as stone, bone or wood. Moreover, it may be helpful to integrate our results regarding the whole 'chaine opératoire' of the bone breakage. Indeed, breakage function could be directed at grease rendering, bone meal, bone fuel or craft activity instead of marrow recover alone. From a comparative perspective, the results of this experiment should be compared with those of numerous ethnographic studies that have observed the butchery practices of current hunter-gatherer peoples. These parallels can include both gestures and cultural practices related to the marrow recovery, by focusing on the difficulties encountered in bone breakage leading to the discarding of a particular bone, by linking the ratio of gain to loss of energy (i.e.: [1, 6, 20, 23, 75–77]). In addition, experiment performed with experts in long bone breakage will be completed this present study and its results. This study is part of a larger project based on the long bone breakage that allowed testing further hypotheses. Despite the fact that our results look promising, they are still preliminary and it still a lot needs to be done.

Percussion marks are very instructive in an archaeological context to access past subsistence behaviour. Our results show that percussion traces on fossil bones are only the tip of the iceberg. They only reveal a small window into past bone marrow extraction practices. This experiment highlighted the variability of experimenters' behaviour and the different bone responses during fracturing in order to propose new hypotheses on past butchering practices based on the comparison of the different data analysed in this work.

To conclude, the influence of bone morphology on the distribution of percussion marks in the archaeological record should be taken into account before inferring cultural traditions. It is essential to compare the intuitive patterns highlighted by spatial analysis with the patterns identified in archaeological levels. It could also prove interesting in archaeological contexts to consider carefully percussion mark distribution on the femur as this could be the most instructive bone element, in terms of marrow recovery methods and group specificity.

## Supporting information

**S1 Appendix.**
(DOCX)

## Acknowledgments

We thank the Museum national d'Histoire naturelle to permit the experiment and P. Meritte for his help. Many thanks to E. Pellé and Z. Thalaud of the Service de Préparation Ostéologique et Taxidermique (SPOT) from the Muséum national d'Histoire naturelle in Paris, for the osteological preparation of all the long bones. We also wish to thank L. Demay for her precious advice during proofreading and her valuable help during the experiment. L. Crépin and colleagues provided helping logistical advice and assistance during the experiment. We are deeply grateful to the volunteers for their selfless efforts in helping with our experiment. We also thank L. Albessard, J. Duveau and G. Mauran for their precious advice. L. Byrne, an official translator and native English speaker, edited the English manuscript.

## Author Contributions

**Conceptualization:** Delphine Vettese.

**Data curation:** Delphine Vettese, Trajanka Stavrova, Juan Marín.

**Formal analysis:** Delphine Vettese, Antony Borel.

**Funding acquisition:** Delphine Vettese, Camille Daujeard.

**Investigation:** Delphine Vettese, Trajanka Stavrova, Juan Marín.

**Methodology:** Delphine Vettese, Trajanka Stavrova, Antony Borel.

**Project administration:** Delphine Vettese.

**Resources:** Delphine Vettese.

**Software:** Delphine Vettese, Antony Borel.

**Supervision:** Delphine Vettese.

**Validation:** Antony Borel, Marie-Hélène Moncel, Marta Arzarello, Camille Daujeard.

**Visualization:** Delphine Vettese, Trajanka Stavrova, Antony Borel.

**Writing – original draft:** Delphine Vettese, Trajanka Stavrova.

**Writing – review & editing:** Delphine Vettese, Trajanka Stavrova, Antony Borel, Juan Marín, Marie-Hélène Moncel, Marta Arzarello, Camille Daujeard.

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
