## [Decision Letter · Decision Letter 0]

15 Apr 2021

PONE-D-21-06808

A way to break bones? The weight of intuitiveness

PLOS ONE

Dear Dr. Vettese,

Thank you for submitting your manuscript to PLOS ONE. After careful consideration, we feel that it has merit but does not fully meet PLOS ONE’s publication criteria as it currently stands. Therefore, we invite you to submit a revised version of the manuscript that addresses the points raised during the review process.

The manuscript "A way to break bones? The weight of intuitiveness" is within the frame of experimental archaeology and deals with the experimental fracturing of long bones in order to assess breakage methods. The paper successfully improves knowledge on this topic and it is suitable for publication. Nevertheless, I think that the suggestions from both the reviewers could greatly help to improve the quality of the paper and its relevance for the scientific community. I loo forward to receive an update version of this work.

We look forward to receiving your revised manuscript.

Kind regards,

Enza Elena Spinapolice, Ph.D

Academic Editor

PLOS ONE

Journal Requirements:

2. In your manuscript, please provide additional information regarding the specimens used in your study. Ensure that you have reported specimen numbers and complete repository information, including museum name and geographic location.

For more information on PLOS ONE's requirements for paleontology and archaeology research, see https://journals.plos.org/plosone/s/submission-guidelines#loc-paleontology-and-archaeology-research.

[We express our gratitude to CHARAL S.A.S. who kindly provided all the bones used in the preparation of this experiment. We are grateful for the funding of the Fondation Nestlé France.]

Please clarify the sources of funding (financial or material support) for your study.

 [This projectwas supported by the Fondation Nestlé

France (SJ 671–16)  (https://fondation.nestle.fr/); the Centre

d’Information des Viandes – Viande, sciences et société (SJ 334–17);

and the Muséum national d’Histoire naturelle. The funders had no role in study design, data collection and analysis, decision to publish, or preparation of the manuscript.]

4. Thank you for stating the following in the Financial Disclosure section:

[This projectwas supported by the Fondation Nestlé

France (SJ 671–16)  (https://fondation.nestle.fr/); the Centre

d’Information des Viandes – Viande, sciences et société (SJ 334–17);

and the Muséum national d’Histoire naturelle. The funders had no role in study design, data collection and analysis, decision to publish, or preparation of the manuscript.]. 

We note that you received funding from a commercial source: Nestlé France

5. Please include captions for all your Supporting Information files at the end of your manuscript, and update any in-text citations to match accordingly. Please see our Supporting Information guidelines for more information: http://journals.plos.org/plosone/s/supporting-information.

6. We note that Supplementary Figure 1 includes an image of an individual.

Reviewers' comments:

Reviewer's Responses to Questions

**Comments to the Author**

1. Is the manuscript technically sound, and do the data support the conclusions?

Reviewer #1: Partly

Reviewer #2: Yes

2. Has the statistical analysis been performed appropriately and rigorously? 

Reviewer #1: Yes

Reviewer #2: Yes

3. Have the authors made all data underlying the findings in their manuscript fully available?

Reviewer #1: Yes

Reviewer #2: Yes

4. Is the manuscript presented in an intelligible fashion and written in standard English?

Reviewer #1: Yes

Reviewer #2: Yes

5. Review Comments to the Author

Reviewer #1: The paper by Vettese et al. deals with the experimental fracturing of long bones in order to assess breakage methods and patterns within what the authors define an “intuitive context”.

Although the overall design of the experiments and the results are sound, my concerns are precisely regarding the idea of “intuitiveness”.

The authors do not explicitly explain what they mean with this term (maybe they should have done so), but from what is reported in lines 20-21 that counter-intuitive means culturally influenced, we may infer that intuitive means NOT culturally influenced.

I may agree that an unexpected and repeated “anomaly” in the breakage pattern may be evidence for a cultural influence, but in a Paleolithic context where the aim is to obtain the maximum amount of marrow with the minimum effort for survival, maybe your “culture” is just asking you to optimize the process, therefore physical constraints (e.g, the shape of the bones, as also evidenced in this study) may be more relevant in selecting the breakage method for the sake of efficiency.

Although the selection of unexperienced experimenters may be a possible approach, I would expect that in the past none in the community would have been completely without previous experience, at least by simply observing other people breaking bones. Maybe some searches in the ethnographic literature of this “learning by observation” would be interesting.

In the INTRODUCTION several times there is an unnecessary focus on Neanderthals and the Middle Palaeolithic, although marrow extraction has been very important in previous periods as well as in the Upper Palaeolithic; furthermore, the authors themselves at the end of this paper make comparisons also with the later period and contemporary Nunamiut, moreover the experimenters are obviously H. sapiens. Therefore, in this introductory part the authors should refer more generally to the Paleolithic adding some references relevant at least to the later period, since these are also discussed in the article.

Lines 68-69 “The definition of a butchery tradition is a systematic and counterintuitive pattern shared by a same group”; as mentioned before why should a tradition be necessarily counterintuitive? Although a repeated “anomaly” in the breakage pattern may represent a tradition (i.e., a cultural trait overcoming common sense), if the aim is the optimization of marrow extraction the pattern recorded may follow common wisdom even within a tradition.

From the information in the “MATERIAL” section it appears that although probably the general results will not change much, the data presented are only a first step of a much larger and needed experimental work:

- Each experimenter broke only one type of skeletal element, therefore there is NO control over his/her behavior with other bones.

- Although the authors explain why they did not use metapodials in this experiment, this bone is often used by both past and present populations for marrow extraction. Therefore, an important bone is missing.

- There is NO control over the effects of handedness on all the skeletal elements selected; only individuals 5 & 6 with femur and tibia where lefthanded.

- There is NO control over the effects of freezing and thawing on all the skeletal elements selected. Furthermore, the authors do not discuss the effects of freezing and thawing on the breakage of the two frozen series of Femurs and Tibiae (nos, 11 and 12); even if there was no difference this should have been explicitly mentioned in the paper.

Therefore, there is a need for further experiments to validate the results.

Line 93 Although it can be seen from the pictures in the Supporting information, the paper should probably mention explicitly the kind of “ground” of the designated area.

Lines 104-105 The shape of the hammerstone and the anvil should also be mentioned.

In Table 2 the knapping knowledge should also be added.

Line 123 The experience of the observer/s should also be mentioned. Is the observer different from the person who analyzed the bones? This further character should be in case introduced.

Line 187-188 The proposed “Efficiency Index calculated by dividing the mass of marrow extracted from each bone by the number of blows”, does not take into account the whole amount of marrow available in the bone that is of course dependent on the skeletal element (as also indicated by the authors later in the paper), maybe using in the index the ratio between the marrow extracted and the marrow available in that skeletal element instead of just the simple marrow extracted may help to overcome this problem.

Line 214 the ambidextrous individual is not reported in table 2.

Line 214 when you say “both hands” do you mean using alternatively one hand or the other, or grasping the hammerstone with two hands?

Line 215 “both hands” has the same meaning as above or different?

Line 218 “medulla” is “medullary cavity”?

Table 3 “Humérus” and “Total général” are still in French; in the version of the Table I received there are some formatting problems (e.g., numbers with decimals are on two lines), make sure that this does not happen in the final version.

Lines 241-242 As mentioned above for the Efficiency index the marrow extracted is not taking into account the total amount of marrow available (in theory) for that element, so you cannot just compare two different skeletal elements.

Line 153 and following; on the evaluation of the quality of marrow maybe add something about the evolution within the same series. Judgments for each skeletal element are based only on 3 individuals!

Table 5 should be reorganized: in the text you talk about quality first and then difficulty, while in the table difficulty comes first and then quality.

Lines 303-305 As commented above the total quantity of obtainable marrow is element dependent therefore comparisons between elements should not be made with this index as it is. Try to modify the index and see if results are the same,

Lines 338-339 The fact that a different observer may have been responsible for observed inter-individual differences is a significant problem, meaning that observation is not objective enough and the results of this experiment cannot be replicated. Furthermore, this major problem is not sufficiently discussed in this paper.

Line 480 “could applied” = could apply?/ applied?

Lines 481-482 The fact that some of the experimenters where afraid of hurting themselves is probably culturally dependent so the supposed “intuitive context” is not so granted after all.

In line 483 you state that “tiredness resulted in a reduction of the force applied”, but in lines 278-79 you say also that “exhaustion from the activity did not influence whether individuals experienced more difficulty at the end of the breakage series”; the concepts are not exactly the same, however, in the end fatigue did have some influence on the bone breakage activity and this should be explored further.

Lines 563-565 “..for two-thirds of the individual bone elements, the number of blows is higher than the number of percussion marks and more surprisingly, for one-third of the bone elements, the number of percussion marks is higher than the number of blows.” Although I understand your point, as it is written, from a simple mathematical point of view, it is obvious that if for 2/3 a>b, for the remaining 1/3 b>a, why should be the second one more surprising?

Lines 570-71 In the explanation provided for the higher number of percussion marks vs. blows that “The use of an anvil or the hammerstone could explain the differences recorded between the number of marks and blows”, I may understand the role of the anvil in producing more percussion marks, but it is not clear to me what features of the hammerstone may increase the number of marks. Maybe explain more explicitly both.

Line 584 “However, we sometimes recorded more marks than blows.” Do you mean “However, in this case too we sometimes recorded more marks than blows”. ?

Line 586 As mentioned before the “shape of the hammerstone” and the type of ground should have been described in the materials & methods section.

Line 634 The ethnographic comparisons should be implemented. The Nunamiut are not the only population breaking bones, there are other possible examples (e.g., Hadza) living in different climatic conditions.

Line 685-86 Since the patterns are largely dependent on the morphology of the skeletal element (something that is outside the human being) they cannot be defined as “intuitive”. Humans are easily and rapidly able to assess (by reasoning) the morphological constraints of a bone and adapt their behavior accordingly in order to be as efficient as possible.

Line 697 For the future experiments I would suggest adding cooking time and heating of the bone w/out meat before breakage to facilitate marrow extraction (as done for example by Nunamiuts).

Reviewer #2: Manuscript number: PONE-D-21-06808

Title: A way to break bones? The weight of intuitiveness

Corresponding author: Delphine Vettese

A fundamental part in archaeological research is based on experimentation and actualism. The present manuscript is within this framework, trying to build upon previous studies focused on experimental series on bone percussion and its relationship with the archaeological record. In my opinion, this manuscript is a very interesting methodological contribution and apart from some issues discussed below (calling for small changes, reorganizations and additions) is suitable for publication.

The first sentence of the abstract should be explained and supported with references in the main text. Fat nutrients are an important nutritional value regardless of the season of the year, and as far as I know, there is still no archaeological study linking seasonality with marrow consumption. I understand that the authors say this on the basis of ethnographic parallels, but the statement should be properly supported and explained in detail in the Introduction section. I think it is a very interesting discussion issue.

I really liked the Introduction on the importance of fat, but I think this topic should also be extended, especially the ethnographic parallels where different ways to break and preserve marrow are described depending on the strategy of obtaining the animals. I think that a little more extension in this regard would enrich the state-of-the-art.

I think that, in the Methods, the main variables to control should be more clearly described at the beginning of the section: state of the bone at the moment of breaking (frozen bones vs fresh bones?), experience, etc. Be careful with the state of the bone since it could also lead to different breakage planes (e.g., higher proportion of mixed angles, irregular planes, etc.).

In addition, it would be good to specify that only one technique has been used to break open the bones (only hammer stone?). This is important to nuance because various techniques could lead not only to different patterns but also to different types of percussion notches. For example, the complete removal of the periosteum resulted in a notch pattern with a larger diameter in the experimentation carried out by Assaf et al (2020). The total removal of the periosteum before breaking allowed the expansion of the impact following the collagen lines and in consequence a larger notches.

Assaf E, Caricola I, Gopher A, Rosell J, Blasco R, Bar O, et al. (2020) Shaped stone balls were used for bone marrow extraction at Lower Paleolithic Qesem Cave, Israel. PLoS ONE 15(4): e0230972.

I liked the idea of proposing an Efficiency Index (EFI) to assess the relationship between blows and marrow for each anatomical element. That is interesting because it also gives us insight into the most difficult bones to break. Most ethnographic accounts report radio-ulna as the most difficult element to fracture, so much so that some groups (for example, the! Kung people; see studies conducted by Yellen) discard it -that is, it is not worth the effort invested in its fracture for the amount of marrow it contains.

Perhaps it would be good to move some of the sentences from the "Data recorded during the experiment" section to the supplementary material, especially the descriptions of the individuals positions, etc. This would make it easier to read and follow the study with the main data highlighted in the main text. But I understand that this is the decision of the authors.

As minor editing changes. Lines 94-96: change the place of Table 2 in the sentence, maybe at the end?

I can only congratulate the authors for an interesting study and their efforts in trying to find cultural patterns in the faunal record through bone breakage.

6. PLOS authors have the option to publish the peer review history of their article (what does this mean?). If published, this will include your full peer review and any attached files.

Reviewer #1: No

Reviewer #2: No

---

## [Author Response · Author response to Decision Letter 0]

17 Jun 2021

Journal Requirements:

 We have made the changes. We have added the new affiliations of two co-authors.

2. In your manuscript, please provide additional information regarding the specimens used in your study. Ensure that you have reported specimen numbers and complete repository information, including museum name and geographic location.

For more information on PLOS ONE's requirements for paleontology and archaeology research, see https://journals.plos.org/plosone/s/submission-guidelines#loc-paleontology-and-archaeology-research.

 We have made the changes.

[We express our gratitude to CHARAL S.A.S. who kindly provided all the bones used in the preparation of this experiment. We are grateful for the funding of the Fondation Nestlé France.]

Please clarify the sources of funding (financial or material support) for your study.

 [This projectwas supported by the Fondation Nestlé

France (SJ 671–16) (https://fondation.nestle.fr/); the Centre

d’Information des Viandes – Viande, sciences et société (SJ 334–17);

and the Muséum national d’Histoire naturelle. The funders had no role in study design, data collection and analysis, decision to publish, or preparation of the manuscript.]

We have made the changes.

Please include your amended statements within your cover letter; we will change the online 

4. Thank you for stating the following in the Financial Disclosure section:

[This project was supported by the Fondation Nestlé

France (SJ 671–16) (https://fondation.nestle.fr/); the Centre

d’Information des Viandes – Viande, sciences et société (SJ 334–17);

and the Muséum national d’Histoire naturelle. The funders had no role in study design, data collection and analysis, decision to publish, or preparation of the manuscript.]. 

We note that you received funding from a commercial source: Nestlé France

The Fondation Nestlé France is not the company Nestlé France is non-profit foundation. Therefore, there are not competing interest. 

5. Please include captions for all your Supporting Information files at the end of your manuscript, and update any in-text citations to match accordingly. Please see our Supporting Information guidelines for more information: http://journals.plos.org/plosone/s/supporting-information.

 We have made the changes.

6. We note that Supplementary Figure 1 includes an image of an individual.

 We join the Plos One consent form. 

Reviewers' comments:

Reviewer's Responses to Questions

Comments to the Author

1. Is the manuscript technically sound, and do the data support the conclusions?

Reviewer #1: Partly

Reviewer #2: Yes

2. Has the statistical analysis been performed appropriately and rigorously? 

Reviewer #1: Yes

Reviewer #2: Yes

3. Have the authors made all data underlying the findings in their manuscript fully available?

Reviewer #1: Yes

Reviewer #2: Yes

4. Is the manuscript presented in an intelligible fashion and written in standard English?

Reviewer #1: Yes

Reviewer #2: Yes

5. Review Comments to the Author

Reviewer #1: The paper by Vettese et al. deals with the experimental fracturing of long bones in order to assess breakage methods and patterns within what the authors define an “intuitive context”.

Although the overall design of the experiments and the results are sound, my concerns are precisely regarding the idea of “intuitiveness”.

The authors do not explicitly explain what they mean with this term (maybe they should have done so), but from what is reported in lines 20-21 that counter-intuitive means culturally influenced, we may infer that intuitive means NOT culturally influenced.

We have added in the text a precision lines 75-76: “by intuitiveness, in this context, we mean known or perceived by intuition: directly apprehended” and lines-20-21 we deleted “culturally induced”

I may agree that an unexpected and repeated “anomaly” in the breakage pattern may be evidence for a cultural influence, but in a Paleolithic context where the aim is to obtain the maximum amount of marrow with the minimum effort for survival, maybe your “culture” is just asking you to optimize the process, therefore physical constraints (e.g, the shape of the bones, as also evidenced in this study) may be more relevant in selecting the breakage method for the sake of efficiency.

Although the selection of unexperienced experimenters may be a possible approach, I would expect that in the past none in the community would have been completely without previous experience, at least by simply observing other people breaking bones. Maybe some searches in the ethnographic literature of this “learning by observation” would be interesting.

In the INTRODUCTION several times there is an unnecessary focus on Neanderthals and the Middle Palaeolithic, although marrow extraction has been very important in previous periods as well as in the Upper Palaeolithic; furthermore, the authors themselves at the end of this paper make comparisons also with the later period and contemporary Nunamiut, moreover the experimenters are obviously H. sapiens. Therefore, in this introductory part the authors should refer more generally to the Paleolithic adding some references relevant at least to the later period, since these are also discussed in the article.

In the introduction, we did a reference to Neandertals because the first paper of Blasco et al. 2013 focus on this question (highlighting butchering traditions) dealt with Neandertal sites. Moreover, we included only two references on the introduction. However, we have added some references of Upper Palaeolithic sites to enrich the introduction line 55. 

Lines 68-69 “The definition of a butchery tradition is a systematic and counterintuitive pattern shared by a same group”; as mentioned before why should a tradition be necessarily counterintuitive? Although a repeated “anomaly” in the breakage pattern may represent a tradition (i.e., a cultural trait overcoming common sense), if the aim is the optimization of marrow extraction the pattern recorded may follow common wisdom even within a tradition.

In absolute terms, we agree with the reviewer, but in the case of a transmitted systematic practice similar to an intuitive one, it would not be possible to distinguish it from a non-transmitted intuitive practice. So the question of the tradition as a practice in the Paleolithic can only be evidenced by differentiating it from the intuitive one. That is why we propose this definition of butchering traditions on Palaeolithic context.

From the information in the “MATERIAL” section it appears that although probably the general results will not change much, the data presented are only a first step of a much larger and needed experimental work:

- Each experimenter broke only one type of skeletal element, therefore there is NO control over his/her behavior with other bones.

Indeed, in this study, only a part of the experimental material provided in 2017 is used. The whole material has not yet been studied, with some experimenters fracturing two different bones. Other experimental studies were performed , with experimenters fracturing each of the four skeletal elements, but that is not the focus of this study. Here, we selected 12 individuals who each broke one bone, without any previous experience in bone breakage. 

- Although the authors explain why they did not use metapodials in this experiment, this bone is often used by both past and present populations for marrow extraction. Therefore, an important bone is missing.

Yes, it is right. It will be improved when we could obtain those types of bone elements complete, which is quite difficult in our context. Indeed, in the case of large-scale butchery waste, these parts of the animal are usually mechanically destroyed. In any case, Capaldo and Blumenschine (1994) indicate that, due to their thicker cortical, metapodials are not suitable bones to study breakage features, as the fractures patterns are non-diagnostic.

- There is NO control over the effects of handedness on all the skeletal elements selected; only individuals 5 & 6 with femur and tibia where lefthanded.

Yes, there are not. However, regarding our sample, it does not influence the pattern observed. 

- There is NO control over the effects of freezing and thawing on all the skeletal elements selected. Furthermore, the authors do not discuss the effects of freezing and thawing on the breakage of the two frozen series of Femurs and Tibiae (nos, 11 and 12); even if there was no difference this should have been explicitly mentioned in the paper.

We have added a sentence, lines 615-616: “Our results show that, there is not impact of freeze on the percussion marks distribution. Moreover, during the experiment all the bones were completely defrosted.”

Therefore, there is a need for further experiments to validate the results.

Here, we present the results of 120 broken long bones. We are fully aware that the results presented here represent a preliminary research and are only focused on some parameters. In preparation of future research, we have already performed additional experimentations in order to test some variables highlighted in this work. We plan to publish these complementary results. 

Line 93 Although it can be seen from the pictures in the Supporting information, the paper should probably mention explicitly the kind of “ground” of the designated area.

We have added brackets line 101: “(outside, earthen soil)”

Lines 104-105 The shape of the hammerstone and the anvil should also be mentioned.

We have added line 113: “a limestone quadrangular anvil” to describe the shape of the anvil. However, we have mentioned that the pebbles are not modified, so they are round without sharp edges.

In Table 2 the knapping knowledge should also be added.

We have added in legend of the table 2: “only the individual n°9 has knapping knowledge”.

Line 123 The experience of the observer/s should also be mentioned. Is the observer different from the person who analyzed the bones? This further character should be in case introduced

The observers are different people with a basic knowledge of bone breakage. It was different from the people who analyzed the bones. However, all the breakage sessions were filmed, and the searcher who studied the bone remains watched them carefully to complete when it was necessary the data recorded during the experiments. 

Line 187-188 The proposed “Efficiency Index calculated by dividing the mass of marrow extracted from each bone by the number of blows”, does not take into account the whole amount of marrow available in the bone that is of course dependent on the skeletal element (as also indicated by the authors later in the paper), maybe using in the index the ratio between the marrow extracted and the marrow available in that skeletal element instead of just the simple marrow extracted may help to overcome this problem.

On this point, we would like to invite the reviewer to refer to a first report published online on PCI Archeology (see below). 

“We are aware of this difference between the elements tested. However, we were able to observe among the different elements proposed to the same individual, there was a difference in size, in the quantity of spongiosa in the bone that could lead to a difference in the quantity of marrow. It was very complex without prior treatment to know the exact marrow content of each bone. For these reasons, we consciously choose to test the empirical hypothesis of whether the fact of having intrinsically less accessible marrow (as for the radius, where the marrow is contained in the proximal diaphysis for example) leads to a multiplication of the number of blows, of remains produced. It is, of course, possible to look at the amount of marrow extracted on average for each type of element, and to normalize it by an average capacity. However, we can observe that there is on average more marrow contained in the femur, but it is not the bone from which the most marrow has been extracted. And we think that is an important issue to explore further. While recalling that the experimenters were novices, and some before their first bone, they did not really know where the marrow was and therefore where to place the blows.”

Line 214 the ambidextrous individual is not reported in table 2.

We have added this sentence, lines 222-223: “That means that the individual n°2 used only his right hand only and the n°5 used simultaneously both hands to hit bones”. 

Line 214 when you say “both hands” do you mean using alternatively one hand or the other, or grasping the hammerstone with two hands?

We mean two hands, we have added: “grasping the hammerstone with two hands all the time.” 

Line 215 “both hands” has the same meaning as above or different?

It is the same meaning, we added line 224: “simultaneously”. 

Line 218 “medulla” is “medullary cavity”?

Yes, it is. We did the modification.

Table 3 “Humérus” and “Total général” are still in French; in the version of the Table I received there are some formatting problems (e.g., numbers with decimals are on two lines), make sure that this does not happen in the final version.

We have corrected the words in the Table 3.

Lines 241-242 As mentioned above for the Efficiency index the marrow extracted is not taking into account the total amount of marrow available (in theory) for that element, so you cannot just compare two different skeletal elements.

Our purpose is to take into account the marrow recovered by each individual independently of the marrow amount inside each bone. We have observed a variation of bones sizes, therefore of the marrow quantity, regarding each skeletal element. 

Line 153 and following; on the evaluation of the quality of marrow maybe add something about the evolution within the same series. Judgments for each skeletal element are based only on 3 individuals!

We did not observe any change in the evaluation of the quality of the extracted marrow for any of the 12 individuals (following table, from the raw data provided in supplementary information). The criteria chosen are not influenced by the skeletal element. 

Table 5 should be reorganized: in the text you talk about quality first and then difficulty, while in the table difficulty comes first and then quality.

We have made the change.

Lines 303-305 As commented above the total quantity of obtainable marrow is element dependent therefore comparisons between elements should not be made with this index as it is. Try to modify the index and see if results are the same,

We propose the index independently of the skeletal element for the reasons exposed in the previous comments.

Lines 338-339 The fact that a different observer may have been responsible for observed inter-individual differences is a significant problem, meaning that observation is not objective enough and the results of this experiment cannot be replicated. Furthermore, this major problem is not sufficiently discussed in this paper.

The two researchers, who analyzed percussion marks, worked together. They applied the same protocol regarding the mark recording process; one of them checked the work of the other one. We have added two sentences: “However, both researchers applied the same protocol and each checked their records.”

Line 480 “could applied” = could apply?/ applied?

We have made the change.

Lines 481-482 The fact that some of the experimenters where afraid of hurting themselves is probably culturally dependent so the supposed “intuitive context” is not so granted after all.

The worry of injury could be due to a total lack of knowledge of the activity (bone breakage), the gestures and their amplitude. In addition, a lack of knowledge of the tools (hammerstone) and of the bone. Considering this, it would seem that this is entirely part of the intuitive nature of the task.

In line 483 you state that “tiredness resulted in a reduction of the force applied”, but in lines 278-79 you say also that “exhaustion from the activity did not influence whether individuals experienced more difficulty at the end of the breakage series”; the concepts are not exactly the same, however, in the end fatigue did have some influence on the bone breakage activity and this should be explored further.

On the one hand, the tiredness can influence the force that is applied at a given moment, and be different between the first bone and the tenth; on the other hand, the feeling of difficulty throughout the experiment can evolve differently because it is influenced by other factors: 

- a better knowledge of the bone and of the location hit,

- the manner of hitting, 

- the force applied, 

- the experience acquired, 

- the bone itself, 

- the cortical thickness

- the bone fragilities...

The following figure (from the raw data we provide as supplementary information) shows that, changes in the evaluation of degree of difficulty feeling for any of the 12 individuals

We made this change line 499 regarding the comment: “at the end of the experiment, tiredness could result in a reduction of the force applied”.

Lines 563-565 “..for two-thirds of the individual bone elements, the number of blows is higher than the number of percussion marks and more surprisingly, for one-third of the bone elements, the number of percussion marks is higher than the number of blows.” Although I understand your point, as it is written, from a simple mathematical point of view, it is obvious that if for 2/3 a>b, for the remaining 1/3 b>a, why should be the second one more surprising?

Yes, it is. The surprise do not come from the count, but from the fact, the blow do not every time create a percussion mark, so the multiplicity of the percussion marks are not directly linked to the impact point. The counter blow due to the anvil could create percussion marks and one blow could create more than one percussion mark due to the irregularity of the hammerstone even if it is not modified. This need to be tested further. 

Lines 570-71 In the explanation provided for the higher number of percussion marks vs. blows that “The use of an anvil or the hammerstone could explain the differences recorded between the number of marks and blows”, I may understand the role of the anvil in producing more percussion marks, but it is not clear to me what features of the hammerstone may increase the number of marks. Maybe explain more explicitly both.

Regarding the anvil, the contact with it produces a counterblow, that could produce an additional percussion marks; the shape of the hammerstone which is round but could have irregularities, could also produce many pits.

Line 584 “However, we sometimes recorded more marks than blows.” Do you mean “However, in this case too we sometimes recorded more marks than blows”. ?

Yes, we have changed the sentence. 

Line 586 As mentioned before the “shape of the hammerstone” and the type of ground should have been described in the materials & methods section.

We have made some additional changes, as we mention in previous comments.

Line 634 The ethnographic comparisons should be implemented. The Nunamiut are not the only population breaking bones, there are other possible examples (e.g., Hadza) living in different climatic conditions.

Yes, it is, however, we only found information on the exact location of the percussion marks within the Binford ethnoarchaeological data (Binford 1981). In this paper, we focus on the percussion marks distribution, and it is the only ethnographic analysis to record it systematically. We have made these following changes: “His observations describe systematic breakage with the recording of the location of the percussion marks after hitting always the same bone portions”. Line 655-656. However, the Nunamiut study is the only one to systematically record the location of the percussion marks

Line 685-86 Since the patterns are largely dependent on the morphology of the skeletal element (something that is outside the human being) they cannot be defined as “intuitive”. Humans are easily and rapidly able to assess (by reasoning) the morphological constraints of a bone and adapt their behavior accordingly in order to be as efficient as possible.

By intuitiveness, in this context, we mean known or perceived by intuition: directly apprehended. Therefore, the morphology is a constraint “directly apprehend”, thus, intuitive. The adaptations and changes are not intuitive; it could be to reduce the number of blows or to repeat a way learned, for example. The efficiency could be complicated to evaluate because it could be culturally induced and depends on the definition of the efficiency. 

Line 697 For the future experiments I would suggest adding cooking time and heating of the bone w/out meat before breakage to facilitate marrow extraction (as done for example by Nunamiuts).

The perspectives of heating or boiled are already considered. We are also aware of additional problems as Blasco and colleagues (2020) did, testing the influence of bone storage and breakage after a period of bone abandonment.

Reviewer #2: Manuscript number: PONE-D-21-06808

Title: A way to break bones? The weight of intuitiveness

Corresponding author: Delphine Vettese

A fundamental part in archaeological research is based on experimentation and actualism. The present manuscript is within this framework, trying to build upon previous studies focused on experimental series on bone percussion and its relationship with the archaeological record. In my opinion, this manuscript is a very interesting methodological contribution and apart from some issues discussed below (calling for small changes, reorganizations and additions) is suitable for publication.

The first sentence of the abstract should be explained and supported with references in the main text. Fat nutrients are an important nutritional value regardless of the season of the year, and as far as I know, there is still no archaeological study linking seasonality with marrow consumption. I understand that the authors say this on the basis of ethnographic parallels, but the statement should be properly supported and explained in detail in the Introduction section. I think it is a very interesting discussion issue.

We have added some references to support the first sentence of the Introduction. Furthermore, we have added some archaeological references on the importance of animal lipids within the food resources. Regarding the nutritional value depending of the season of the year, there are few archaeological studies linking seasonality with marrow consumption (i.e. Daujeard 2008, Costamagno 1999 and Speth 1987). However, in perspectives, the comparison of this work with archaeological sites, which occupation seasonality is known, could bring complementary information. 

1. Speth JD, Spielman KA. Energy source, protein metabolism, and hunter-gatherer subsistence strategies. J Anthropol Archaeol. 1983;2 (1): 1–31. 

2. Hardy K, Brand-miller J, Brown KD, Thomas MG, Copeland L. The Importance of Dietary Carbohydrate in Human Evolution. Quaterly Rev Biol. 2006;90 (3): 251–268. 

3. Morin E. Fat composition and Nunamiut decision-making: a new look at the marrow and bone grease indices. J Archaeol Sci. 2007;34: 69–82. doi:10.1016/j.jas.2006.03.015

4. Outram AK. A Comparison of Paleo-Eskimo and Medieval Norse Bone Fat Exploitation in Western Greenland. Artic Anthropol. 1999;36: 103–117. 

5. Cordain L, Watkins BA, Florant GL, Kelher M, Rogers L, Li Y. Fatty acid analysis of wild ruminant tissues: Evolutionary implications for reducing diet-related chronic disease. Eur J Clin Nutr. 2002;56: 181–191. doi:10.1038/sj.ejcn.1601307

13. Morin E, Soulier M-C. New Criteria for the Archaeological Identification of Bone Grease Processing. Am Antiq. 2017;82: 96–122. doi:10.1017/aaq.2016.16

14. Thompson JC, Carvalho S, Marean CW, Alemseged Z. Origins of the Human Predatory Pattern: The Transition to Large-Animal Exploitation by Early Hominins. Curr Anthropol. 2019;60: 1–23. doi:10.1086/701477

15. Outram AK. Distinguishing bone fat exploitation from other taphonomic processes: what caused the high level of bone fragmentation at the Middle Neolithic site of Ajvide, Gotland? In: Mulville J, Outram AK, editors. The zooarchaeology of fats, oils, milk and dairying. Oxford: Oxbow Books; 2005. pp. 32–43

I really liked the Introduction on the importance of fat, but I think this topic should also be extended, especially the ethnographic parallels where different ways to break and preserve marrow are described depending on the strategy of obtaining the animals. I think that a little more extension in this regard would enrich the state-of-the-art.

We have add this sentence with references: “As current population of hunter-gatherer living in cold and dry environment are characterized by hyperproteinic diets [1,6,8,16].” Lines 50-51.

Kelly RL. The Lifeways of Hunter-Gatherers: The Foraging Spectrum. University. Cambridge; 2013.

I think that, in the Methods, the main variables to control should be more clearly described at the beginning of the section: state of the bone at the moment of breaking (frozen bones vs fresh bones?), experience, etc. Be careful with the state of the bone since it could also lead to different breakage planes (e.g., higher proportion of mixed angles, irregular planes, etc.).

Yes, this is the case for the fracture planes. However, there does not seem to have any role on the distribution of impact marks and the percussion marks typology. 

In addition, it would be good to specify that only one technique has been used to break open the bones (only hammer stone?). This is important to nuance because various techniques could lead not only to different patterns but also to different types of percussion notches. For example, the complete removal of the periosteum resulted in a notch pattern with a larger diameter in the experimentation carried out by Assaf et al (2020). The total removal of the periosteum before breaking allowed the expansion of the impact following the collagen lines and in consequence a larger notches.

Assaf E, Caricola I, Gopher A, Rosell J, Blasco R, Bar O, et al. (2020) Shaped stone balls were used for bone marrow extraction at Lower Paleolithic Qesem Cave, Israel. PLoS ONE 15(4): e0230972.

Lines 114 and 608, we specified: “The periosteum was not removed before breakage”. Moreover, the table 3 describes the different techniques used by each experimenter. The detailed information was indicated in the Supplementary information 14. One of our results is the convergence of the percussion mark distribution pattern even with the use of different techniques. We will publish another work using the morphometric data recorded on this experiment. 

I liked the idea of proposing an Efficiency Index (EFI) to assess the relationship between blows and marrow for each anatomical element. That is interesting because it also gives us insight into the most difficult bones to break. Most ethnographic accounts report radio-ulna as the most difficult element to fracture, so much so that some groups (for example, the! Kung people; see studies conducted by Yellen) discard it -that is, it is not worth the effort invested in its fracture for the amount of marrow it contains.

We have added some sentences to explain the future comparison of our results with ethnographical ones: lines 727 – 732 with some references, “From a comparative perspective, the results of this experiment should be compared with those of numerous ethnographic studies that have observed the butchery practices of current hunter-gatherer peoples. These parallels can include both gestures and cultural practices related to the marrow recovery, by focusing on the difficulties encountered in bone breakage leading to the discarding of a particular bone, by linking the ratio of gain to loss of energy (i.e.: [1,8,11,14,66,67]).” However, we think that the amount of ethnographic data prevent an the appropriate comparison in this paper. 

Perhaps it would be good to move some of the sentences from the "Data recorded during the experiment" section to the supplementary material, especially the descriptions of the individuals positions, etc. This would make it easier to read and follow the study with the main data highlighted in the main text. But I understand that this is the decision of the authors.

We have chosen to keep this part in the main text, because we would like to highlight the variation of the individual and bone position in relation to the percussion marks distribution, being described for the first time.

As minor editing changes. Lines 94-96: change the place of Table 2 in the sentence, maybe at the end?

We have made the change.

I can only congratulate the authors for an interesting study and their efforts in trying to find cultural patterns in the faunal record through bone breakage.

We thank the useful comments of the second reviewer, which enrich our paper. 

6. PLOS authors have the option to publish the peer review history of their article (what does this mean?). If published, this will include your full peer review and any attached files.

Do you want your identity to be public for this peer review? For information about this choice, including consent withdrawal, please see our Privacy Policy.

Reviewer #1: No

Reviewer #2: No

---

## [Decision Letter · Decision Letter 1]

27 Jul 2021

PONE-D-21-06808R1

A way to break bones? The weight of intuitiveness

PLOS ONE

Dear Dr. Vettese,

Thank you for submitting your manuscript to PLOS ONE. After careful consideration, we feel that it has merit but does not fully meet PLOS ONE’s publication criteria as it currently stands. Therefore, we invite you to submit a revised version of the manuscript that addresses the points raised during the review process.

We look forward to receiving your revised manuscript.

Kind regards,

Enza Elena Spinapolice, Ph.D

Academic Editor

PLOS ONE

Journal Requirements:

Additional Editor Comments (if provided):

Reviewers' comments:

Reviewer's Responses to Questions

**Comments to the Author**

1. If the authors have adequately addressed your comments raised in a previous round of review and you feel that this manuscript is now acceptable for publication, you may indicate that here to bypass the “Comments to the Author” section, enter your conflict of interest statement in the “Confidential to Editor” section, and submit your "Accept" recommendation.

Reviewer #1: (No Response)

Reviewer #2: All comments have been addressed

2. Is the manuscript technically sound, and do the data support the conclusions?

Reviewer #1: Partly

Reviewer #2: Yes

3. Has the statistical analysis been performed appropriately and rigorously? 

Reviewer #1: Yes

Reviewer #2: Yes

4. Have the authors made all data underlying the findings in their manuscript fully available?

Reviewer #1: Yes

Reviewer #2: Yes

5. Is the manuscript presented in an intelligible fashion and written in standard English?

Reviewer #1: Yes

Reviewer #2: Yes

6. Review Comments to the Author

Reviewer #1: Since the previous review round many of the questions have been addressed in this new version of the paper by Vettese et al., however, in some of the answers there are explanations only for the reviewer, but then actual changes or more explicit clarifications were not included in the paper.

Maybe something should be added to the text for these points too.

1) For example (NB line numbers, here as elsewhere, refer to the first version):

Reviewer:

Line 123 The experience of the observer/s should also be mentioned. Is the observer different from the person who analyzed the bones? This further character should be in case introduced

Authors:

The observers are different people with a basic knowledge of bone breakage. It was different from the people who analyzed the bones. However, all the breakage sessions were filmed, and the searcher who studied the bone remains watched them carefully to complete when it was necessary the data recorded during the experiments.

Reviewer:

Line 187-188 The proposed “Efficiency Index calculated by dividing the mass of marrow extracted from each bone by the number of blows”, does not take into account the whole amount of marrow available in the bone that is of course dependent on the skeletal element (as also indicated by the authors later in the paper), maybe using in the index the ratio between the marrow extracted and the marrow available in that skeletal element instead of just the simple marrow extracted may help to overcome this problem.

Authors:

On this point, we would like to invite the reviewer to refer to a first report published online on PCI Archeology (see below).

“We are aware of this difference between the elements tested. However, we were able to observe among the different elements proposed to the same individual, there was a difference in size, in the quantity of spongiosa in the bone that could lead to a difference in the quantity of marrow. It was very complex without prior treatment to know the exact marrow content of each bone. For these reasons, we consciously choose to test the empirical hypothesis of whether the fact of intrinsically less accessible marrow having (as for the radius, where the marrow is contained in the proximal diaphysis for example) leads to a multiplication of the number of blows, of remains produced. It is, of course, possible to look at the amount of marrow extracted on average for each type of element, and to normalize it by an average capacity. However, we can observe that there is on average more marrow contained in the femur, but it is not the bone from which the most marrow has been extracted. And we think that is an important issue to explore further. While recalling that the experimenters were novices, and some before their first bone, they did not really know where the marrow was and therefore where to place the blows.”

Reviewer:

Lines 563-565 “..for two-thirds of the individual bone elements, the number of blows is higher than the number of percussion marks and more surprisingly, for one-third of the bone elements, the number of percussion marks is higher than the number of blows.” Although I understand your point, as it is written, from a simple mathematical point of view, it is obvious that if for 2/3 a>b, for the remaining 1/3 b>a, why should be the second one more surprising?

Authors:

Yes, it is. The surprise do not come from the count, but from the fact, the blow do not every time create a percussion mark, so the multiplicity of the percussion marks are not directly linked to the impact point. The counter blow due to the anvil could create percussion marks and one blow could create more than one percussion mark due to the irregularity of the hammerstone even if it is not modified. This need to be tested further.

Reviewer:

Lines 570-71 In the explanation provided for the higher number of percussion marks vs. blows that “The use of an anvil or the hammerstone could explain the differences recorded between the number of marks and blows”, I may understand the role of the anvil in producing more percussion marks, but it is not clear to me what features of the hammerstone may increase the number of marks. Maybe explain more explicitly both.

Authors:

Regarding the anvil, the contact with it produces a counterblow, that could produce an additional percussion marks; the shape of the hammerstone which is round but could have irregularities, could also produce many pits.

2) In the first line of the abstract the authors should refer more generally to the Palaeolithic and not only to the Middle Palaeolithic following the corrections already made in the main text.

3) To the reviewer first comment on the sentence:

Lines 68-69 “The definition of a butchery tradition is a systematic and counterintuitive pattern shared by a same group”; as mentioned before why should a tradition be necessarily counterintuitive? Although a repeated “anomaly” in the breakage pattern may represent a tradition (i.e., a cultural trait overcoming common sense), if the aim is the optimization of marrow extraction the pattern recorded may follow common wisdom even within a tradition

The authors answer :“In absolute terms, we agree with the reviewer, but in the case of a transmitted systematic practice similar to an intuitive one, it would not be possible to distinguish it from a non-transmitted intuitive practice. So the question of the tradition as a practice in the Paleolithic can only be evidenced by differentiating it from the intuitive one. That is why we propose this definition of butchering traditions on Palaeolithic context”

However, this means that it is only possible to evidence “cultural anomalies”, and this is of course true in general, but one should not ignore, just because it is not possible to see them clearly, that “normal” cultural patterns following what the authors would consider as the result of intuition do exist and may be probably related to strategies for the optimization of resource acquisition. Furthermore, these “normal” cultural patterns may be the majority.

This topic needs to be discussed in te text.

4) The paper does not evidence enough the fact that this is only a preliminary study, part of a larger project. Furthermore, some statements (e.g., lack of influence of freezing/thawing on breakage) are based only on few cases, nevertheless the authors appear (too) sure of their findings.

The awareness that sill a lot needs to be done should appear more clearly in the text and the authors should explain why they decided to submit only this part of their project (i.e., why they consider this set of experiments as complete on their own).

At this preliminary stage of the project maybe more nuanced statements would be more appropriate.

5) Possibly if this project (and consequently the paper) included also “experienced bone breakers” the proposed results, if confirmed, would have had a more solid background.

6) The fact that the bones were defrosted when broken does not necessarily imply that there had been no effects, at a microscopic level, on bone structure produced by ice crystals, especially larger ones like those resulting from “homemade” (vs. industrial) freezing.

In the end I still have problems with the concept of intuitiveness and for me "directly apprehended" is something different from intuition.

Probably, this “intuitiveness” concept adopted by the authors may be similar to the one employed for example for a new computer program or a new cell phone that may be immediately very easy to use, but still, even in this case, just consider that what may be intuitive and easy for a young person, more familiar with electronic devices, may not be so for a more aged individual; therefore, also “intuitiveness” may in many cases be culturally influenced.

Reviewer #2: I am grateful to the authors for their effort in modifying the text to address the main weaknesses.

I think that in general the critical points have been solved, although there is still a minor critical point about the variables to control (especially, the state of the bone at the moment of breaking [e.g., frozen bones vs fresh bones?]) to solve. The authors in their answer state that the state of the bone does not affect the distribution of percussion marks ; however I think this statement (and a short explanation) should be clearly described in the text to make it clear, as it can be a contentious point when reading the study. If this clarification is added to the manuscript, I think it is ready for publication.

Thanks to the authors for this nice contribution.

7. PLOS authors have the option to publish the peer review history of their article (what does this mean?). If published, this will include your full peer review and any attached files.

Reviewer #1: No

Reviewer #2: No

---

## [Author Response · Author response to Decision Letter 1]

5 Aug 2021

6. Review Comments to the Author

Reviewer #1: Since the previous review round many of the questions have been addressed in this new version of the paper by Vettese et al., however, in some of the answers there are explanations only for the reviewer, but then actual changes or more explicit clarifications were not included in the paper.

Maybe something should be added to the text for these points too.

1) For example (NB line numbers, here as elsewhere, refer to the first version):

Reviewer:

Line 123 The experience of the observer/s should also be mentioned. Is the observer different from the person who analyzed the bones? This further character should be in case introduced

Authors:

The observers are different people with a basic knowledge of bone breakage. It was different from the people who analyzed the bones. However, all the breakage sessions were filmed, and the searcher who studied the bone remains watched them carefully to complete when it was necessary the data recorded during the experiments.

Following the suggestion of the reviewer, we have added these sentences.

Reviewer:

Line 187-188 The proposed “Efficiency Index calculated by dividing the mass of marrow extracted from each bone by the number of blows”, does not take into account the whole amount of marrow available in the bone that is of course dependent on the skeletal element (as also indicated by the authors later in the paper), maybe using in the index the ratio between the marrow extracted and the marrow available in that skeletal element instead of just the simple marrow extracted may help to overcome this problem.

Authors:

On this point, we would like to invite the reviewer to refer to a first report published online on PCI Archeology (see below).

“We are aware of this difference between the elements tested. However, we were able to observe among the different elements proposed to the same individual, there was a difference in size, in the quantity of spongiosa in the bone that could lead to a difference in the quantity of marrow. It was very complex without prior treatment to know the exact marrow content of each bone. For these reasons, we consciously choose to test the empirical hypothesis of whether the fact of intrinsically less accessible marrow having (as for the radius, where the marrow is contained in the proximal diaphysis for example) leads to a multiplication of the number of blows, of remains produced. It is, of course, possible to look at the amount of marrow extracted on average for each type of element, and to normalize it by an average capacity. However, we can observe that there is on average more marrow contained in the femur, but it is not the bone from which the most marrow has been extracted. And we think that is an important issue to explore further. While recalling that the experimenters were novices, and some before their first bone, they did not really know where the marrow was and therefore where to place the blows.”

We have made this change. “Despite this difference between the elements tested, we observe differences between the bones even in the same individual series: there was a difference in size, in the quantity of spongiosa and in the quantity of marrow. The knowledge of the exact marrow content of each bone was complex to determine before the extraction and varied according the bones. For these reasons, we tested if the fact of intrinsically less accessible marrow having (as for the radius, where the marrow is contained in the proximal diaphysis for example) leads to a multiplication of the number of blows and of remains produced. Our results show that, even if there is on average more marrow contained in the femur, but it is not the bone from which the most marrow has been extracted. While recalling that the experimenters were novices, and some before their first bone, they did not really know where the marrow was and therefore where to place the blows.”

Reviewer:

Lines 563-565 “..for two-thirds of the individual bone elements, the number of blows is higher than the number of percussion marks and more surprisingly, for one-third of the bone elements, the number of percussion marks is higher than the number of blows.” Although I understand your point, as it is written, from a simple mathematical point of view, it is obvious that if for 2/3 a>b, for the remaining 1/3 b>a, why should be the second one more surprising?

Authors:

Yes, it is. The surprise do not come from the count, but from the fact, the blow do not every time create a percussion mark, so the multiplicity of the percussion marks are not directly linked to the impact point. The counter blow due to the anvil could create percussion marks and one blow could create more than one percussion mark due to the irregularity of the hammerstone even if it is not modified. This need to be tested further.

We have made this change. “The surprise does not come from the count, but from the fact, the blow do not every time create a percussion mark, so the multiplicity of the percussion marks are not directly linked to the impact point. The counter blow due to the anvil could create percussion marks. Besides, one blow could create more than one percussion mark due to the irregularity of the hammerstone, even if it is not modified. But this need to be further explored.”

Reviewer:

Lines 570-71 In the explanation provided for the higher number of percussion marks vs. blows that “The use of an anvil or the hammerstone could explain the differences recorded between the number of marks and blows”, I may understand the role of the anvil in producing more percussion marks, but it is not clear to me what features of the hammerstone may increase the number of marks. Maybe explain more explicitly both.

Authors:

Regarding the anvil, the contact with it produces a counterblow, that could produce an additional percussion marks; the shape of the hammerstone which is round but could have irregularities, could also produce many pits.

We have made this change. “Therefore, the use of an anvil could create supplementary marks during the blow that is the counterblow. The shape of the hammerstone, which is round but could have irregularities, could also produce many pits.”

2) In the first line of the abstract the authors should refer more generally to the Palaeolithic and not only to the Middle Palaeolithic following the corrections already made in the main text.

We have made this change.

3) To the reviewer first comment on the sentence:

Lines 68-69 “The definition of a butchery tradition is a systematic and counterintuitive pattern shared by a same group”; as mentioned before why should a tradition be necessarily counterintuitive? Although a repeated “anomaly” in the breakage pattern may represent a tradition (i.e., a cultural trait overcoming common sense), if the aim is the optimization of marrow extraction the pattern recorded may follow common wisdom even within a tradition

The authors answer :“In absolute terms, we agree with the reviewer, but in the case of a transmitted systematic practice similar to an intuitive one, it would not be possible to distinguish it from a non-transmitted intuitive practice. So the question of the tradition as a practice in the Paleolithic can only be evidenced by differentiating it from the intuitive one. That is why we propose this definition of butchering traditions on Palaeolithic context”

However, this means that it is only possible to evidence “cultural anomalies”, and this is of course true in general, but one should not ignore, just because it is not possible to see them clearly, that “normal” cultural patterns following what the authors would consider as the result of intuition do exist and may be probably related to strategies for the optimization of resource acquisition. Furthermore, these “normal” cultural patterns may be the majority.

This topic needs to be discussed in the text.

We have made this change. We agree with the reviewer that this could be discussed. Nevertheless, taking into account our current knowledge, it is and probably will be almost impossible to know and to discern what would be of the order of “normality” or “cultural anomalies”. That is why, we choose to not discuss this aspect in this present paper. 

4) The paper does not evidence enough the fact that this is only a preliminary study, part of a larger project. Furthermore, some statements (e.g., lack of influence of freezing/thawing on breakage) are based only on few cases, nevertheless the authors appear (too) sure of their findings.

The awareness that sill a lot needs to be done should appear more clearly in the text and the authors should explain why they decided to submit only this part of their project (i.e., why they consider this set of experiments as complete on their own).

At this preliminary stage of the project maybe more nuanced statements would be more appropriate.

We agree with the reviewer and we have made some nuances, notably in the conclusion. And we have added lines 757-760: “This study is part of a larger project based on the long bone breakage that allowed testing further hypotheses. Despite the fact that our results look promising, they are still preliminary and it still a lot needs to be done.”

Regarding the sample chosen, the results previously presented in Stavrova et al. 2019 required us to expand our sample size of the series studied. We chose this number of three series for each of the four elements because they were complete. In addition, these series contained enough bones and traces with different characteristics that allow testing statistically the variables we had previously identified.

5) Possibly if this project (and consequently the paper) included also “experienced bone breakers” the proposed results, if confirmed, would have had a more solid background.

We agree with the reviewer that the experiment with experienced bone breakers need to be test. That is why we performed an experiment with expert experimenters, the results of these will be the subject of further publications which is part of our larger project. The paper is already long and present significant results that should be completed with complementary publication on the subject. We added this statement to clarify lines 757-758: “In addition, experiment performed with experts in long bone breakage will be completed this present study and its results.”

6) The fact that the bones were defrosted when broken does not necessarily imply that there had been no effects, at a microscopic level, on bone structure produced by ice crystals, especially larger ones like those resulting from “homemade” (vs. industrial) freezing.

We agree with the reviewer that why, lines 413-417, we added: “Besides, the bone frozen presented mixed fracture planes [52]. However, there does not seem to have any role on the distribution of impact marks and the percussion marks typology. The 20 elements (10 humerus and 10 radio-ulnas) that have been frozen do not show a divergent distribution pattern compared to those that have not been frozen”.

In the end I still have problems with the concept of intuitiveness and for me "directly apprehended" is something different from intuition.

Probably, this “intuitiveness” concept adopted by the authors may be similar to the one employed for example for a new computer program or a new cell phone that may be immediately very easy to use, but still, even in this case, just consider that what may be intuitive and easy for a young person, more familiar with electronic devices, may not be so for a more aged individual; therefore, also “intuitiveness” may in many cases be culturally influenced.

We understand the reviewer confusion; however we used the definition of the Merriam-Webster dictionary. Moreover, the protocol was made with this concept and to approach an intuitive practice without previous knowledge on the activity of long bone breakage to extract the marrow. We tried all along the paper, to explain and apply this concept. 

Reviewer #2: I am grateful to the authors for their effort in modifying the text to address the main weaknesses.

I think that in general the critical points have been solved, although there is still a minor critical point about the variables to control (especially, the state of the bone at the moment of breaking [e.g., frozen bones vs fresh bones?]) to solve. The authors in their answer state that the state of the bone does not affect the distribution of percussion marks ; however I think this statement (and a short explanation) should be clearly described in the text to make it clear, as it can be a contentious point when reading the study. If this clarification is added to the manuscript, I think it is ready for publication.

Thanks to the authors for this nice contribution.

We agree with the reviewer and this comment is similar with the first reviewer that is why we added line 413-417: “Besides, the bone frozen presented mixed fracture planes [52]. However, there does not seem to have any role on the distribution of impact marks and the percussion marks typology. The 20 elements (10 humerus and 10 radio-ulnas) that have been frozen do not show a divergent distribution pattern compared to those that have not been frozen”.

---

## [Decision Letter · Decision Letter 2]

14 Oct 2021

A way to break bones? The weight of intuitiveness

PONE-D-21-06808R2

Dear Dr. Vettese,

We’re pleased to inform you that your manuscript has been judged scientifically suitable for publication and will be formally accepted for publication once it meets all outstanding technical requirements.

Kind regards,

Enza Elena Spinapolice, Ph.D

Academic Editor

PLOS ONE

**Comments to the Author**

1. If the authors have adequately addressed your comments raised in a previous round of review and you feel that this manuscript is now acceptable for publication, you may indicate that here to bypass the “Comments to the Author” section, enter your conflict of interest statement in the “Confidential to Editor” section, and submit your "Accept" recommendation.

Reviewer #1: All comments have been addressed

---

## [Editor Report · Acceptance letter]

18 Oct 2021

PONE-D-21-06808R2 

A way to break bones? The weight of intuitiveness 

Dear Dr. Vettese:

I'm pleased to inform you that your manuscript has been deemed suitable for publication in PLOS ONE. Congratulations! Your manuscript is now with our production department. 

Kind regards, 

on behalf of

Dr. Enza Elena Spinapolice 

Academic Editor

PLOS ONE